# Tuning electron delocalization of hydrogen-bonded organic framework cathode for high-performance zinc-organic batteries

Wenda Li[1], Hengyue Xu[2], Hongyi Zhang[1], Facai Wei[1], Lingyan Huang[1], Shanzhe Ke[1], Jianwei Fu[3], Chengbin Jing[1], Jiangong Cheng[4] & Shaohua Liu[1] ✉

Stable cathodes with multiple redox-active centres affording a high energy density, fast redox kinetics and a long life are continuous pursuits for aqueous zinc-organic batteries. Here, we achieve a high-performance zinc-organic battery by tuning the electron delocalization within a designed fully conjugated two-dimensional hydrogen-bonded organic framework as a cathode material. Notably, the intermolecular hydrogen bonds endow this framework with a transverse two-dimensional extended stacking network and structural stability, whereas the multiple $C=O$ and $C=N$ electroactive centres cooperatively trigger multielectron redox chemistry with super delocalization, thereby sharply boosting the redox potential, intrinsic electronic conductivity and redox kinetics. Further mechanistic investigations reveal that the fully conjugated molecular configuration enables reversible $Zn^{2+}/H^+$ synergistic storage accompanied by 10-electron transfer. Benefitting from the above synergistic effects, the elaborately tailored organic cathode delivers a reversible capacity of 498.6 mAh $g^{-1}$ at 0.2 A $g^{-1}$, good cyclability and a high energy density (355 Wh $kg^{-1}$).

Because of the inherent merits of resource abundance, high theoretical capacity (820 mAh $g^{-1}$), environmental benignity and safety, rechargeable aqueous zinc-ion batteries are booming for next-generation large-scale energy storage[1,2]. Accordingly, great efforts have been made to explore high-performance cathode materials for zinc-ion batteries[3–5]. In particular, organic electrode molecules as cathode materials have received growing attention in recent years due to their environmental friendliness, sustainability, structural designability, and abundant resources[6]. However, they still suffer from limited electron conductivity, low energy density, and low cyclability, resulting from the intrinsic molecular structure, random stacking properties, and inevitable dissolution of functional groups[7,8]. Consequently, exploring new types of organic cathodes

that can overcome the above barriers is vital for advanced zinc organic batteries.

Conjugated organic molecules featuring large π-electron configurations have attracted considerable attention as organic electrodes owing to their electrical conductivity, structural designability, and material accessibility[9,10]. Noticeably, the electrochemical activity and redox chemistry of the conjugated molecules can be systematically manipulated by molecular tailoring and electron modulation strategies[11–13]. Despite the significant progress in electrochemical reversibility, the reported conjugated systems obtained by molecular modification and conjugate structure extension strategies always involve some functional groups with insufficient redox activity that inevitably reduce the capacity[14,15]. In addition, the weak electron

[1]State Key Laboratory of Precision Spectroscopy; Engineering Research Center of Nanophotonics & Advanced Instrument (Ministry of Education), School of Physics and Electronic Science, East China Normal University, Shanghai 200241, P.R. China. [2]Tsinghua Shenzhen International Graduate School, Tsinghua University, Shenzhen 518055, P.R. China. [3]School of Materials Science and Engineering, Zhengzhou University, 75 Daxue Road, Zhengzhou 450052, P. R. China. [4]State Key Lab of Transducer Technology, Shanghai Institute of Microsystem and Information Technology, Chinese Academy of Sciences, Shanghai 200050, P. R. China. ✉e-mail: shliu@phy.ecnu.edu.cn

delocalization induced by nucleophilic functional groups makes it difficult to obtain a satisfactory output voltage[16]. Meanwhile, these organic small molecules have a strong tendency to dissolve into the electrolyte during the redox process, thereby leading to poor cyclability[17]. Therefore, to address these challenges, exploiting new conjugated organic cathodes through rational molecule design is highly urgent and significant for high-performance zinc-organic batteries (ZOBs).

Herein, we achieve a super-electron-delocalized fully conjugated two-dimensional (2D) hydrogen-bonded organic framework (F-HOF) by manipulating a strong nucleophilic functionalization strategy, i.e., the benzo[a]benzo[7,8]quinoxalino[2,3-i]phenazine-5,6,8,14,15,17-hexane (BBQPH) F-HOF featuring synergistic C = O and C = N electron-withdrawing motifs that alter the intramolecular electron distribution, thus boosting the redox voltage and triggering multielectron storage chemistry for ZOBs. Impressively, the multiple intermolecular hydrogen bonds (C = N···H/C = O···H) combined with the π−π stacking interactions further enhance the structural stability of BBQPH. In addition, the super electron delocalization simultaneously improves the redox potential, intrinsic electronic conductivity, and redox kinetics. As expected, the BBQPH electrodes deliver a high voltage of 1.2 V, a significantly improved capacity (498.6 mAh g$^{-1}$ at 0.2 A g$^{-1}$), corresponding to an energy density of 355 Wh kg$^{-1}$, and an exceptional cycling performance of >1000 cycles at 5.0 A g$^{-1}$ for aqueous ZOBs. Furthermore, ex situ investigations and theoretical simulations clarify the reversible Zn$^{2+}$/H$^+$ synergistic storage mechanism accompanied by 10-electron transfer. Our work will provide new insight for realizing advanced organic materials through rational molecular design towards diverse applications.

## Results

### Theoretical prediction and guidance for molecular design

A high output voltage together with a high capacity contributes to a high energy density of batteries, which is highly desirable for their practical application[18]. On the one hand, the redox potential of an organic molecule is closely related to its highest occupied molecular orbital (HOMO) energy level, and a lower HOMO energy signifies a higher oxidation capability based on molecular orbital theory prediction, resulting in a higher open-circuit voltage ($V_{oc}$) in ZOBs (Fig. 1a)[19]. Generally, the HOMO energy can be lowered by induced electron delocalization of the electron-withdrawing group[20]. On the other hand, the delivered capacity of organic molecules mainly depends on the number of electroactive centers per weight[21]. Increasing the number of active sites and reducing their relative molecular mass by regulating the π-conjugated building units can realize a high-capacity performance. Currently, rationally designing low-HOMO-energy conjugated molecules with multiple redox-active centers is supposed to realize high-energy-density ZOBs. Given this, a super-electron-delocalized fully conjugated organic molecule, benzo[a]benzo[7,8]quinoxalino[2,3-i]phenazine-5,6,8,14,15,17-hexane (BBQPH), possessing multiple electron-withdrawing carbonyl groups (C = O), was designed (Fig. 1b). Similarly, benzo[a]benzo[7,8]quinoxalino[2,3-i]phenazine-8,17-dione (BBQPD) was also designed as a conceptual reference. Notably, the designed BBQPH intrinsically possesses almost the highest theoretical capacity level (567 mAh g$^{-1}$) compared with the other organic cathode materials previously reported[16].

To precisely verify the electron delocalization of the designed molecules, the electrostatic potential (ESP) and molecular orbitals were first calculated to evaluate the intrinsic electronic properties.

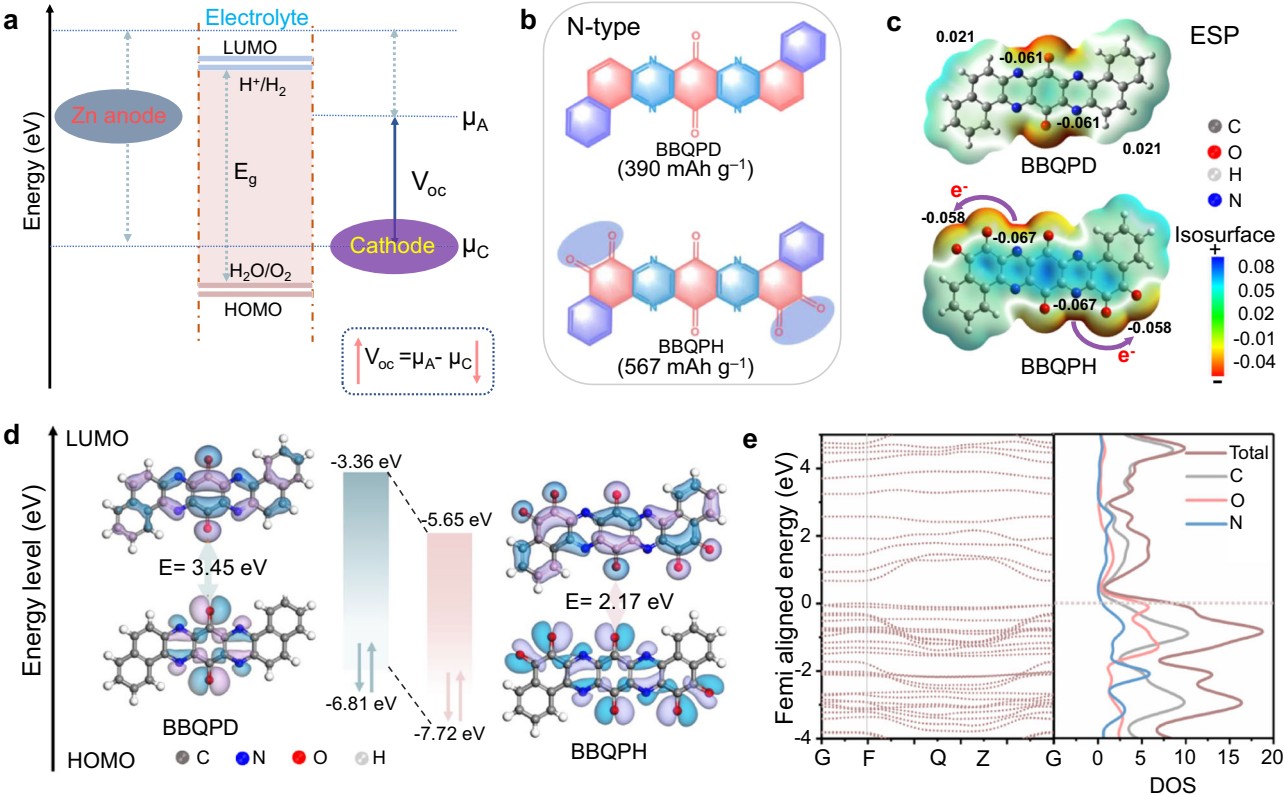

**Fig. 1 | Molecular design based on theoretical prediction and guidance.**
**a** Scheme of the composition of the electrode potentials ($\mu_A$ and $\mu_C$), which are related to the electron energy of molecular orbitals. The open-circuit voltage for ZOBs depends on the HOMO energy of the organic cathode: $V_{oc} = \mu_A - \mu_C$.

**b** Chemical structures of BBQPD and BBQPH. **c** Simulated ESP distributions of BBQPD and BBQPH. **d** Calculated relative HOMO/LUMO energy levels and energy gaps used in the DFT method. **e** Energy band spectrum and corresponding pDOS of the simulated BBQPH.

Compared to BBQPD (0.021 a.u.), the ESP value (−0.058 a.u.) of o-quinone functional groups for BBQPH is dramatically decreased because of its super electron delocalization and highly conjugated symmetric structure[22]. Meanwhile, BBQPH shows more dramatic electron delocalization, which results in smaller molecular dipole moments and robust intramolecular electron migration from the conjugate planar center to the o-quinone groups (Fig. 1c). Moreover, the lower HOMO value of the BBQPH molecule (−7.72 eV) compared with BBQPD (−6.81 eV) indicates a higher cation affinity and promisingly leads to an increase in $V_{oc}$. In addition, the super-electron-delocalized fully conjugated BBQPH demonstrates a smaller gap ($E_g$) (2.17 eV) between the LUMO and HOMO, which endows it with higher electrical conductivity for rapid electron transport as well as fast redox kinetics. Interestingly, there is a significant density of electron states at the Fermi level for the designed BBQPH according to the energy band and partial density of states (pDOS; Fig. 1e and Supplementary Fig. 1), which is primarily contributed by C=O group, verifying that o-quinone increases the super electron delocalization and further underscores the potential for fast charge transfer of BBQPH. Therefore, the elaborately tailored BBQPH would be significantly attractive for ZOBs.

## Synthesis and structural characterizations

BBQPH featuring multiple carbonyls and pyrazine redox sites was synthesized via a two-step solvothermal reaction (Supplementary Figs. 2 and 3). Meanwhile, BBQPD was also synthesized as a reference (Supplementary Fig. 4). The chemical compositions of the samples were investigated using a variety of characterization methods. The [1]H NMR spectra together with high-resolution mass spectra verify the successful synthesis of BBQPH (Supplementary Fig. 5). The Fourier transform infrared (FTIR) spectrum of BBQPH shows peaks located at 1590 and 1689 cm$^{-1}$, which are attributed to the C=N and C=O stretching vibration groups, respectively[23]. The Raman spectra further confirm that the molecular functional structures of the two products are as expected (Supplementary Fig. 6)[24]. Notably, the BBQPH molecule shows a rotationally symmetric molecular configuration consisting of multiple carbonyls and pyrazine groups, which could easily form a fully conjugated 2D hydrogen-bonded organic framework via intermolecular noncovalent locking. Consequently, powder X-ray diffraction (PXRD) was performed to reveal the crystallinity and molecular stacking patterns (Fig. 2a). The prominent diffraction peak at $2\theta = 27.1°$

can be assigned to the (002) plane, which conforms to the simulated AA stacking model, and the interlayer distance between fully conjugated BBQPH layers is 3.3 Å (Fig. 2b). The diffraction peaks at 3.5° and 16.8° correspond to the (100) and (300) crystal facets, respectively. However, BBQPD only exhibits weak amorphous diffraction peaks due to its random molecular arrangement (Supplementary Fig. 7).

To further reveal the stacking orientation of BBQPH, Pawley refinement was performed with an optimized model to fit the obtained experimental data. The refined crystal structure for BBQPH was assigned to the monoclinic system with the P1 space group (Supplementary Table 2), and the unit cell parameters were refined to $a = 17.204$ Å, $b = 9.163$ Å, $c = 3.553$ Å, $\alpha = 114°$, $\gamma = 63.4°$, and $\beta = 94.4°$. The calculated hydrogen bond lengths between the hydrogen atom of C-H and the O/N atoms of the carbonyl/phenazine groups are ~2.35–2.40 Å. Specifically, owing to the centrosymmetric fully conjugated structure with abundant C = O and C = N electron-withdrawing groups, each BBQPH molecule is locked with four adjacent BBQPH molecules by multiple hydrogen bonds between strong hydrogen bond acceptor C = N/C = O functional groups and weak hydrogen donor C-H groups (C = N···H/C = O···H), forming a 2D hydrogen-bonded organic framework structure (Fig. 1b and Supplementary Fig. 8). The high-resolution TEM image (Supplementary Fig. 9) visualizes the ordered and robust structure of the BBQPH materials with a lattice fringe of 0.33 nm, which is consistent with the XRD measurement. Notably, the calculated green peak located at sign($I_2$)r values of −0.02–0.00 in the reduced density gradient diagram (Fig. 2c) further clarifies the existence of π-π interactions between adjacent molecular layers, which agrees well with the PXRD result[25]. It should be noted that the existence of strong intermolecular forces of π-π interactions and hydrogen bonding renders high thermostability and poor solubility in water solvents (Supplementary Figs. 10 and 11), which is significantly advantageous for suppressing the shuttle effect and for a favorable cycling stability in aqueous batteries[26,27].

The porous-honeycomb morphology was revealed by scanning electron microscopy (SEM) (Supplementary Fig. 12) for the prepared BBQPH, which has a larger surface area of 68.12 m$^2$ g$^{-1}$ and a micropore size of 0.55 nm (Supplementary Fig. 13), thus facilitating ion diffusion and shortening the transport pathways as an electrode material. Meanwhile, the modulated BBQPH crystal is verified to have a parallelepiped structure, well matching the simulated molecular model

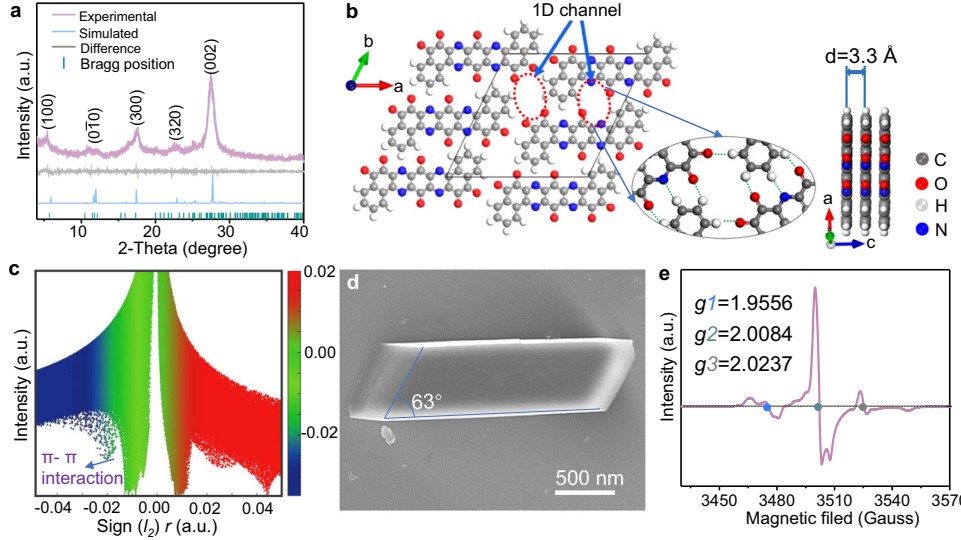

**Fig. 2 | Structural characterization. a** PXRD pattern of the BBQPH framework. **b** Stacking diagram with a packing distance of 3.3 Å together with C = O···H/C = N···H noncovalent locking models of the BBQPH framework. **c** Plot of the density gradient decreasing with sign($\lambda_2$)ρ. **d** SEM image of the modulated BBQPH crystal. **e** EPR spectrum of BBQPH.

(Fig. 2d). The corresponding energy-dispersive spectroscopy (EDS) mapping results further show homogeneous distributions of C, O, and N elements in the modulated BBQPH crystal (Supplementary Fig. 14). Noticeably, the electron paramagnetic resonance spectrum (Fig. 2e) of BBQPH presents three obvious signals with $g$ values of 1.9556, 2.0084 and 2.0237, revealing a typical spin-orbit delocalization coupling effect and the existence of multiple radicals with an unpaired electron[28]. The cation radical could generate electron holes for high electron mobility in the fully conjugated structure. Significantly, the delocalization enhances the spin density of the cation radical in the fully conjugated structure, further confirming its good stability. That is, such a super electron delocalization provided by o-quinone groups induced along the fully conjugated configuration is favorable for electron implantation into the organic skeleton with a low energy barrier and ensures high stability of free radicals.

## Electrochemical performance

The unique structure of the 2D hydrogen-bonded organic framework with electron-delocalization-enhanced redox centers of the BBQPH framework inspired us to further investigate its electrochemical properties in aqueous ZOBs. Cyclic voltammetry (CV) curves were recorded to analyze the reduction/oxidation behavior of BBQPH and BBQPD at 0.5 mV s$^{-1}$ (Fig. 3a and Supplementary Fig. 15). The BBQPH electrode displays three distinct pairs of redox peaks at 1.31/1.16, 0.81/0.72, and 0.58/0.45 V, confirming the three-step 10-electron transfer redox reaction of BBQPH derived from the multiple redox-active

centers[29]. The initial five cycles overlap almost completely, indicating stable and highly reversible redox behavior[16]. Impressively, the following galvanostatic charge/discharge test of the two batteries confirms that the introduction of the carbonyl groups induces super π-electron delocalization, thus significantly elevating the redox potential (1.2 V; Fig. 3b), which is the highest output voltage of an n-type material[30]. The enhanced voltage plateau together with multielectron redox significantly contributes to the enhanced capacity (increases by 45%) and energy density (increases by 25% to 355 Wh kg$^{-1}$) of the ZOBs (Fig. 3c). Specifically, the BBQPH electrode shows good rate performance with energy densities and capacities of 498.6, 455.0, 436.8, 422.9, 406.2, and 393.6 mAh g$^{-1}$ at 0.2, 0.5, 1.0, 2.0, 5.0, and 8.0 A g$^{-1}$, respectively (Fig. 3d and Supplementary Figs. 16 and 17). Moreover, when the current density is returned to 0.2 A g$^{-1}$, the capacity recovers to 487.2 mAh g$^{-1}$, revealing the enhanced rate performance and reversibility of the BBQPH electrode. For the BBQPD electrode, the corresponding capacities monotonically decrease from 305.4 to 208.3 mAh g$^{-1}$ as the applied current density increases from 0.2 to 8.0 A g$^{-1}$.

Remarkably, the ZOBs with BBQPH electrodes exhibit a satisfactory performance of up to 1000 cycles under 5.0 A g$^{-1}$ with a 95% capacity retention (Fig. 3e). In contrast, the BBQPD cathode demonstrates capacity fading to 152.6 mAh g$^{-1}$ after 1000 cycles (Supplementary Fig. 18). Notably, BBQPH represents one of the best in terms of comprehensive performance compared to organic-based electrodes in previous related reports (Fig. 3f and Supplementary Table 3)[31]. To further confirm the practicability and superiority of the BBQPH

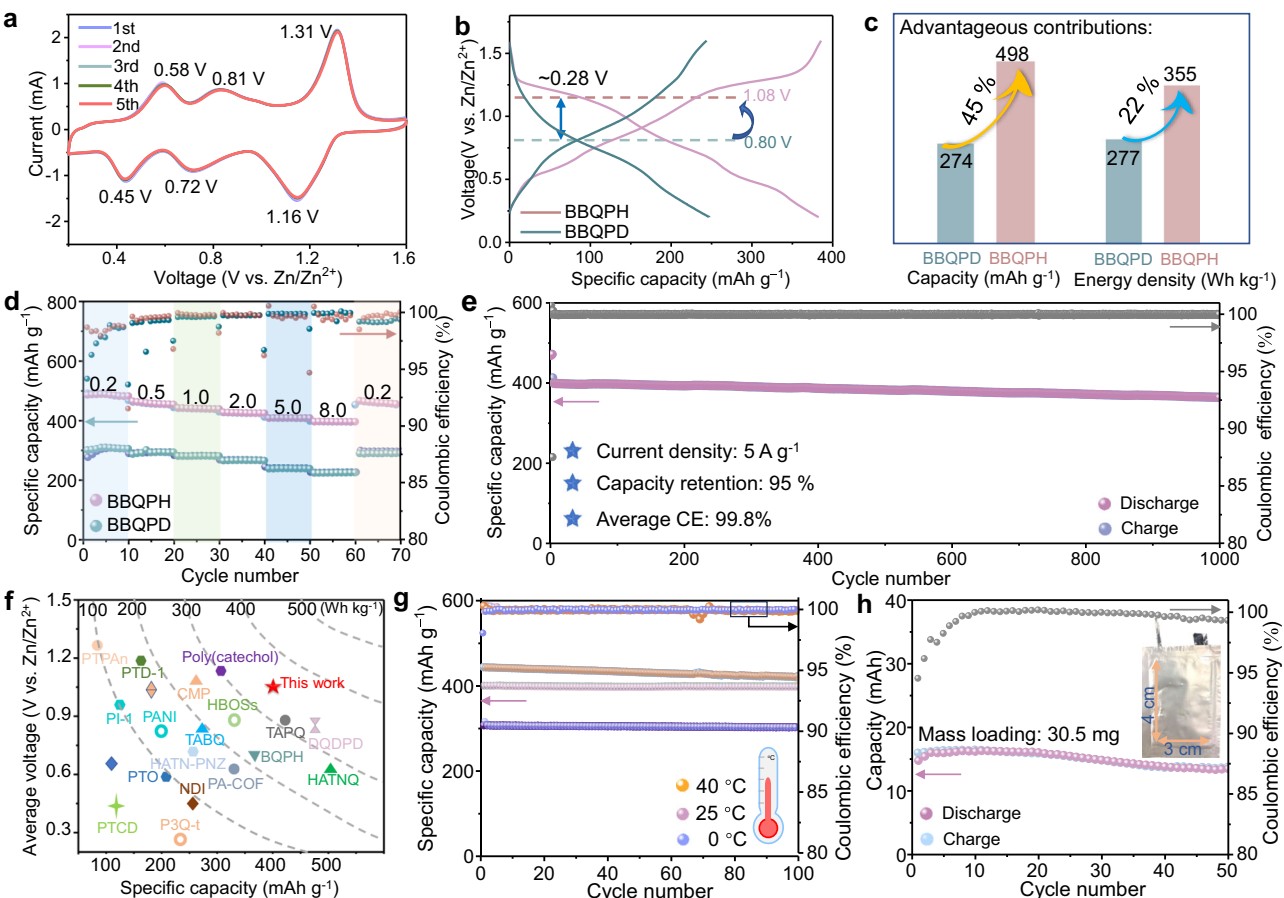

**Fig. 3 | Electrochemical performance of the BBQPH and BBQPD cathodes. a** Contrastive CV curves of the BBQPH and BBQPD cathodes at 0.5 mV s$^{-1}$. **b** Galvanostatic discharge/charge curves of the BBQPH and BBQPD electrodes at 0.2 A g$^{-1}$. **c** Bar graph of the discharge plateau versus the energy density. **d** Rate performances and **e** cycling stability of BBQPH and BBQPD. **f** Comparison of the

capacities and energy densities between the BBQPH electrode and recently reported organic cathodes. **g** Cycling performances of BBQPH electrodes at different test temperatures. **h** Cycling performance of pouch-type Zn//BBQPH batteries at 0.1 A g$^{-1}$.

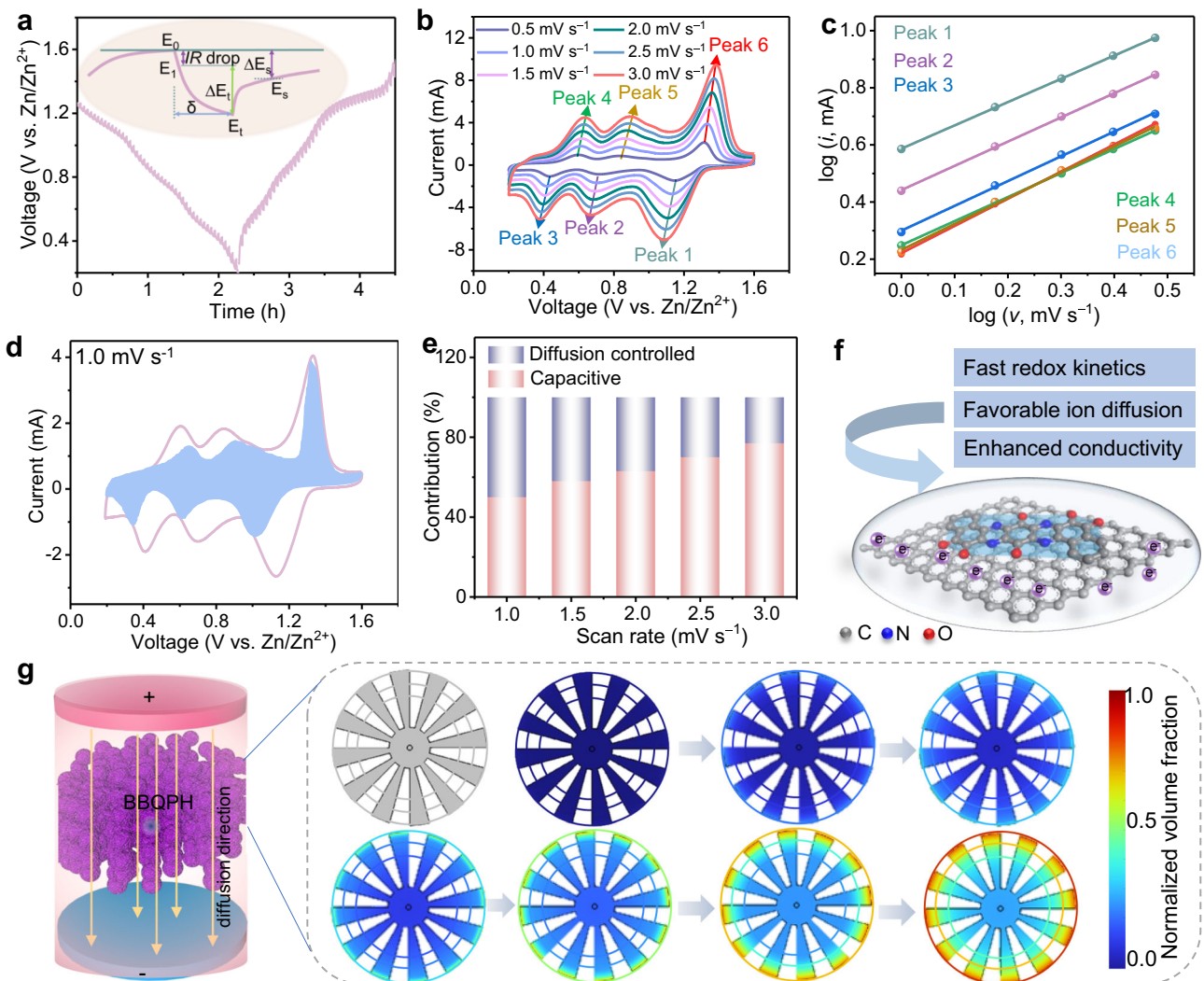

**Fig. 4 | Reaction kinetic analysis. a** GITT curves of the BBQPH cathode at 0.2 mA cm$^{-2}$. **b** CV curves of BBQPH at scan rates ($v$) from 1.0 to 3.0 mV s$^{-1}$. **c** b-values calculated by linearly fitting the peak current ($i$) and scan rate ($v$). **d** Capacitive and diffusion contribution ratios at 1.0 mV s$^{-1}$. **e** Capacity contribution at different scan rates. **f** Schematic of fast ion transport and reaction kinetics of the ZOBs. **g** Computational fluid dynamics simulations for BBQPH based on a 2D model.

electrode, the cycling stability of ZOBs coin cells was evaluated under various working temperatures (Fig. 3g). The BBQPH electrode delivers satisfactory discharge capacities of approximately 409 and 418 mAh g$^{-1}$ and good cycling stability at working temperatures of 25 and 40 °C, respectively. More encouragingly, the battery can still deliver 306 mAh g$^{-1}$ even at 0 °C, revealing a preeminent work tolerance under low temperatures. Meanwhile, the ZOBs can power a timer-integrated display, verifying the promising application prospects in electronic devices (Supplementary Fig. 19). Moreover, the assembled Zn-ion pouch cell outputs a 14.0 mAh capacity at 0.1 A g$^{-1}$ (Fig. 3h and Supplementary Fig. 20), and the exceptional electrochemical performance of the battery holds considerable promise in practical applications and highlights the advantages of this molecular design, including the increased energy density and reduced solubility of organic materials. Overall, combining the abundant electroactive sites and hydrogen-bonding-stabilized network skeleton, the assembled ZOBs show great advantages in terms of sustainability, cycling stability, and energy density as grid-scale energy storage systems.

## Redox kinetics

The galvanostatic intermittent titration technique (GITT) was first carried out to explore the redox kinetics of the two electrodes

(Fig. 4a). The higher ion diffusion coefficient ($1.5 \times 10^{-9}$ cm$^2$ s$^{-1}$ vs. $6.8 \times 10^{-10}$ cm$^2$ s$^{-1}$ for BBQPD) combined with the low activation energy (1.58 eV) calculated from electrochemical reaction curves evidence the faster reaction kinetics and superior ion transport ability of BBQPH (Supplementary Figs. 21 and 22)[32]. To further reveal the origin of the superior kinetic performance of BBQPH, CV tests of the two electrodes were performed at different scan rates (0.5–3.0 mV s$^{-1}$) (Fig. 4b and Supplementary Fig. 15). The BBQPH electrodes present a small peak shift with increasing scan rate, indicating remarkable electrochemical reversibility[33]. Of note, the well-linearly fitted b-value at the redox peaks of the BBQPH electrode varies from 0.69 to 0.78, indicating fast surface-controlled charge-storage kinetics (Fig. 4c)[34]. Almost 86.2% of the total stored charge is contributed by the surface redox reaction at 3.0 mV s$^{-1}$ (Fig. 4d). Considering the surface-controlled charge storage character, the anti-self-discharge and anti-dissolution abilities were determined, further revealing the good capacity retention and structural stability (Supplementary Fig. 23). Accordingly, the dominant reason for the fast charge storage is attributed to strong electron delocalization bringing highly electroactive C=O/C=N zincophilic sites and a robust molecular H-bonded organic network, which facilitates faster redox kinetics, enhanced stability, and elevated conductivity compared to BBQPD (Fig. 4f).

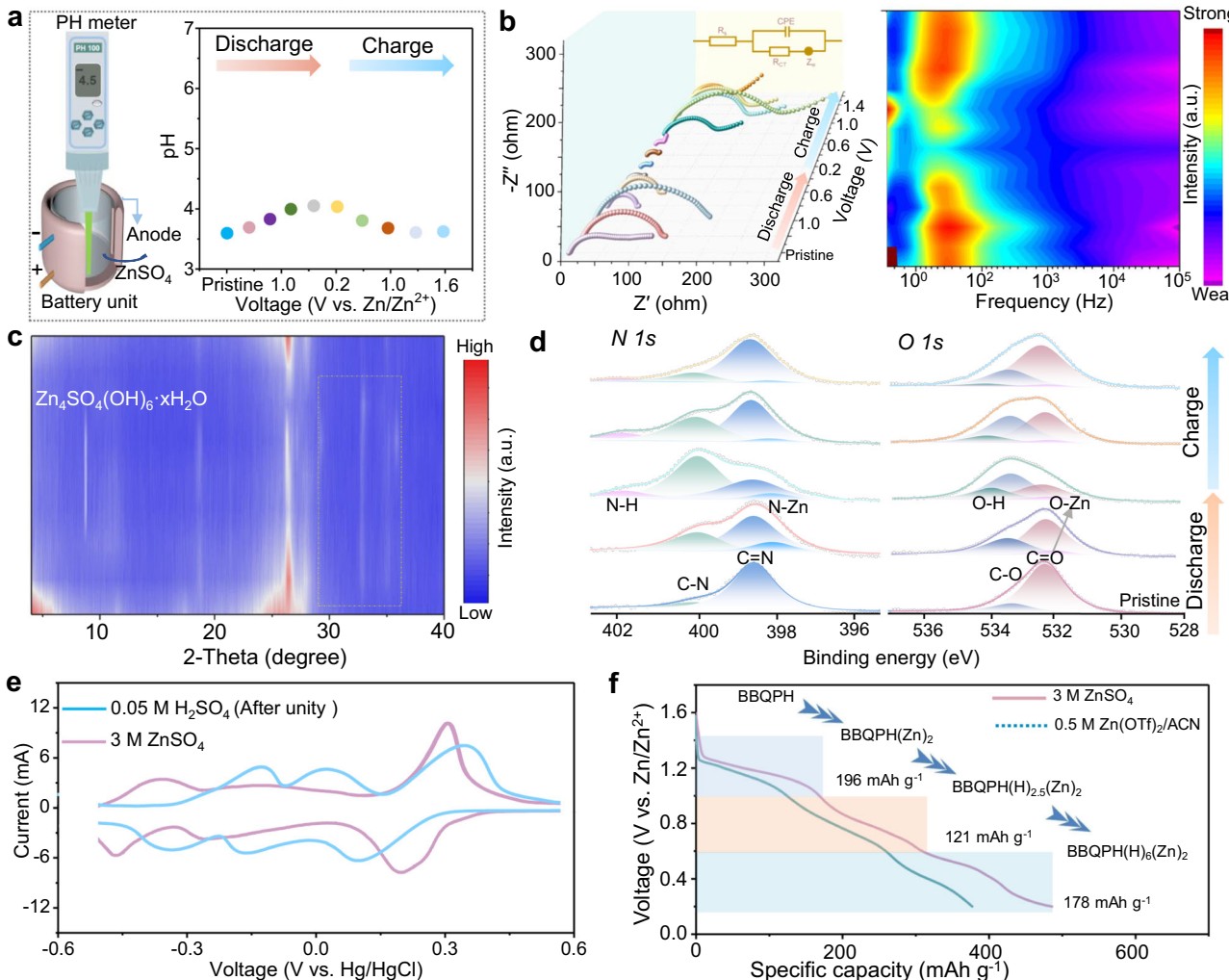

**Fig. 5 | Redox chemistry mechanism. a** Schematic of the in-situ pH detection equipment and corresponding pH evolution diagram of the BBQPH cathode during redox. **b** In-situ EIS spectra and corresponding frequency spectra in first discharge-charge process. **c** Ex-situ high-resolution XPS spectra of *N 1s* and **d** *O 1s*. **e** CV curves of BBQPH at 1.0 mV s⁻¹. **f** Zn²⁺/H⁺ costorage mechanism and possible redox processes of BBQPH at each plateau.

A computational fluid dynamics simulation was also employed to evaluate the effect of the porous-honeycomb structure on electrolyte diffusion[35]. Here, we established a flower-like structure model according to the SEM results to simulate BBQPH (Fig. 4g). Over the course of the simulation, the volume fraction distribution of the fluid (which corresponds to Zn²⁺ in the model) gradually accelerates and eventually reaches 90%, indicating that Zn²⁺ is capable of completely penetrating the deep interior of the model. Additionally, according to the structure model, the normalized volume fraction of ions gradually increases from the surface to the center. The nonporous structure model of BBQPD demonstrates a discontinuous increase in the normalized volume fraction and displays a maximum volume fraction at a radius of ~0.17 from the surface (Supplementary Fig. 24). These simulation results and experimental evidence highlight the tremendous ion diffusion and storage capability of the electron-delocalized fully conjugated H-bonding framework, which further reveals the structural advantages of BBQPH as a cathode of ZOBs.

### Redox chemistry mechanism

We deeply delved into the detailed electrochemical mechanism of BBQPH in ZOBs through a series of characterizations. Note that plentiful protons exist in aqueous Zn²⁺ electrolytes because of the hydrolysis reaction of Zn salt. Therefore, an in-situ pH test was conducted

using a direct-reading pH meter in a homemade two-electrode reaction device to track the pH changes (Fig. 5a). The pH value shows an increase during discharge (voltage range from 1.0 to 0.2 V) and a downward trend when charging, revealing the existence of reversible proton storage behavior for BBQPH[36]. Additionally, in situ EIS was used to monitor the real-time impedance variation, indicating a significant decrease in the charge storage process, which is mainly attributed to the enhanced ion/electron conductivity (Fig. 5b)[37]. Following ex situ XRD, FTIR spectroscopy and X-ray photoelectron spectroscopy (XPS) were further performed to elucidate the chemical structure evolution of BBQPH during the charge storage process. The characteristic peak of Zn₄SO₄(OH)₆·xH₂O (8.17°) derived from proton insertion gradually appears and disappears during discharging and charging, respectively (Fig. 5c and Supplementary Figs. 25 and 26). Moreover, the typical peaks of BBQPH slightly decrease while new peaks appear upon the cation intercalation process, corroborating the Zn²⁺/H⁺ reversible costorage behavior. In addition, the two typical peaks in the FTIR spectra weaken in the reduction process, corresponding to the coordination of cations with redox-active centers (Supplementary Fig. 27). Similarly, the XPS spectra of *O 1s* and *N 1s* reveal the evolution of the C=O and C=N bonds, which are transformed into the C-N bond (400.1 eV) and C-O bond (533.3 eV) upon the discharge process (Fig. 4d), respectively, suggesting the coordination reaction of the two C=N/C=O redox centers with Zn²⁺/H⁺[38,39]. In the subsequent oxidation

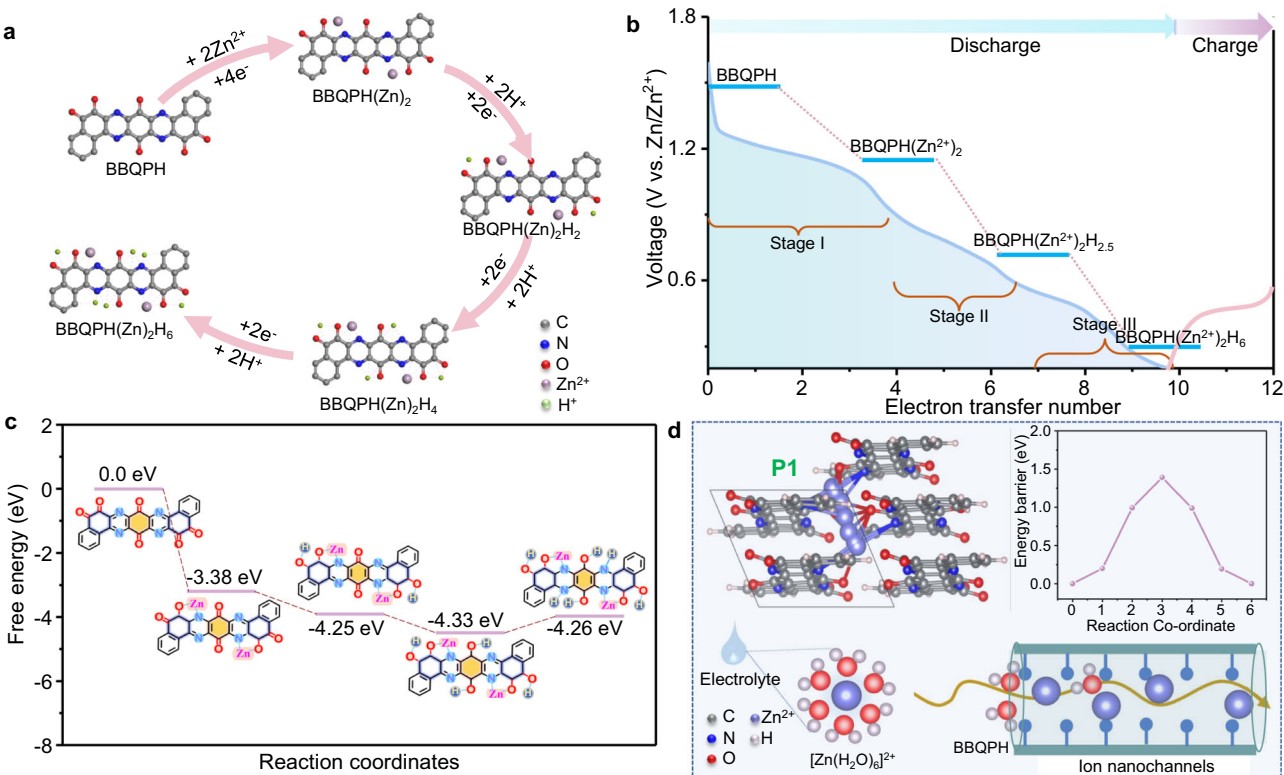

**Fig. 6 | DFT calculations for Zn²⁺/H⁺ synergistic storage in BBQPH. a** Zn²⁺/H⁺ synergistic storage pathway obtained from simulations of BBQPH in the redox process. Calculated potential (**b**) and Gibbs free energy diagrams (**c**) of BBQPH during stepped Zn²⁺/H⁺ coordination. **d** Schematic demonstrating Zn²⁺ migration along the P1 path and the corresponding migration barrier curve.

process, the C=O/C=N peaks reappear and gradually strengthen, illustrating the release of Zn²⁺/H⁺ and the highly reversible feature of the BBQPH redox activity.

To reveal the proton storage contributions, we further investigated the electrochemical properties of the BBQPH electrode using an electrolyte without Zn salt (i.e., 0.05 M H₂SO₄) and aqueous ZnSO₄ electrolytes with different concentrations (Supplementary Figs. 28 and 29). Notably, the BBQPH cathode demonstrates three similar pairs of redox peaks with lower redox potentials in the 0.05 M H₂SO₄ electrolyte according to the calibrated CV curves (Fig. 5e), indicating that the initial coordination is derived from Zn²⁺, followed by two continuous successive H⁺ uptakes. In addition, a nonaqueous Zn²⁺ electrolyte (0.5 M Zn(OTf)₂/ACN) was adopted to further confirm the priority coordination of Zn²⁺ and exclude the effect of protons (Supplementary Figs. 30 and 31), and the higher Zn²⁺-coordination potential further confirms the previous speculation. Interestingly, the BBQPH electrode still exhibits a moderately reduced capacity (372 mAh g⁻¹) but with lower redox kinetics (Supplementary Fig. 32). In view of the above analyses, Zn²⁺ contributes approximately 45% of the capacity in the BBQPH electrode, while the capacity contribution of protons can reach 55%, which is consistent with the EDS mapping analysis and XPS analysis results (Supplementary Fig. 33 and Supplementary Table 1). Based on the above mechanistic studies, it is clear that the carbonyl and phenazine groups of BBQPH have highly reversible redox activity, and the stepwise co-intercalation of Zn²⁺/H⁺ enables BBQPH to exhibit enhanced redox dynamics.

### Theoretical calculation

To clarify the redox path of the 10-electron reduction reactions of BBQPH, we established four cation-coordinated models and quantified the absorption energies of each cation on coordinated structures by theoretical calculation (Supplementary Fig. 34). According to the

calculation results, the stable O···Zn···N coordination (BBQPH(Zn)₂) is first formed by bonding Zn cations with one neighboring O of o-quinone and one N of pyrazine functional groups due to the strongest affinity to the cations[23]. Then, 2H⁺ tends to be separately coordinated with a pair of O atoms of the o-quinone group to form BBQPH(Zn)₂(H)₂. Finally, H⁺ inserts into the phenazine site with two H⁺ bonded by two N atoms, generating BBQPH(Zn)₂(H)₆ and agreeing with the experimental analysis. Based on the above results, the optimized total charge storage path of BBQPH coordinated by Zn²⁺/H⁺ during the multistep redox processes is summarized in Fig. 6a. Moreover, the reduction potentials together with the Gibbs free energy (Fig. 6b, c) were calculated according to the adsorption energy diagram. It is worth mentioning that small energy differences in the adjacent redox process at different steps can lead to overlap of redox peaks. Therefore, at least three main redox potential levels related to the step-by-step 10-electron transfer mechanism were established, in agreement with the CV curve. The Gibbs free energy (ΔG) of the cation coordination models shows a decreasing trend in the first several stages, indicating stable adsorption of cations on the active sites[29]. The increase in ΔG in the final proton intercalation step can be attributed to the deformation of the molecular plane structure. Furthermore, the Zn²⁺ mobility within the BBQPH framework was also computed using the DFT model to determine the optimal ion migration path. Two representative migration paths (P1 and P2) were evaluated based on a transition state search, as shown in Fig. 6d and Supplementary Fig. 35[40]. Specifically, P1 represents the intralayer diffusion of Zn²⁺ ions, while P2 corresponds to Zn²⁺ cation migration along nanochannels. The corresponding diffusion barrier energies for P1 and P2 were calculated to be 10 eV and 1.42 eV, respectively. Accordingly, the lower migration energy barrier within molecularly constructed nanochannels promotes rapid penetration of Zn²⁺/H⁺ to the internal active sites and boosts fast redox dynamics, thus affording an improved rate

performance as well as superb stability during long-term cycling for BBQPH.

## Discussion

In summary, we have demonstrated a super-electron-delocalized fully conjugated 2D hydrogen-bonded organic framework based on molecular engineering for high-energy-density ZOBs. The super electron delocalization alters the intramolecular electron distribution, thus significantly boosting the redox potential. Meanwhile, the intermolecular multiple hydrogen bonds endow the framework with a transverse 2D extension and longitudinal π-π stacking structure, which extends the electron delocalization area, narrows the energy gap between the HOMO and LUMO, and boosts the structural stability. Benefiting from these synergetic superiorities, BBQPH delivers an elevated output voltage, a high reversible capacity of 498.6 mAh g$^{-1}$ at 0.2 A g$^{-1}$, good cyclability (capacity retention of 95% after 1000 cycles), and a high energy density (355 Wh kg$^{-1}$) for ZOBs, indicating that it has advanced prospects in practical applications. Furthermore, the redox kinetics and charge-storage mechanism of BBQPH were systematically investigated, revealing a boosted reversible Zn$^{2+}$/H$^+$ synergistic storage behavior accompanied by 10-electron transfer. This work offers new opportunities for molecular design of redox-reversible multielectron organic cathodes and thus bright practical prospects for high-performance ZOBs.

## Methods
### Materials
All commercially available reagents and solvents were purchased from Adamas, Aladdin, or Innochem Co., Ltd. and used directly without additional purification.

### Synthesis of tetraamino-p-benzoquinone

Tetraamino-p-benzoquinone (TABQ) was prepared based on a simple dehydration condensation reaction together with an oxidation reaction, as shown in Supplementary Fig. 2[1]. In detail, tetrachlorobenzoquinone (TCBQ) (10.0 g, 0.04 mol) and potassium phthalimide (30.0 g, 0.16 mol) were placed in a 250 mL round bottom flask, and 100.0 mL of acetonitrile was slowly added to the above mixture. After stirring at room temperature for 10 min, the solution was further heated and kept at 80 °C for 12 h under stirring. After cooling to room temperature, the suspension was poured into 1.0 L of boiled water, filtered, and washed successively several times. The resulting brown–yellow product was dried at 60 °C overnight to obtain tetra(phthalimido)-benzoquinone (TPBQ; 31.2 g, 78%). The synthesized TPBQ (4.5 g, 0.014 mol) was first dispersed in 100 mL of 80 wt% hydrazine hydrate in a 250 mL three-necked flask under magnetic stirring at room temperature for 2 h. Then, the mixture was heated at 65 °C for 2 h to facilitate the reduction process. Finally, the reaction system was cooled to room temperature, filtered, and washed with water and ethanol, and then, the dark purple product of TABQ was obtained in 45% isolated yield.

### Synthesis of benzo[a]benzo[7,8]quinoxalino[2,3-i]phenazine-8,17-dione (BBQPD) and BBQPH

BBQPH was prepared, inspired separately by the literature, based on a two-step reaction, as shown in Supplementary Fig. 4[41]. Typically, first, 4.5 g (20 mmol) of 2,3-dichloro-1,4-napthoquinone and 1.0 g (10 mmol) of TABQ were added to 150 mL of pyridine solution, and the mixture was stirred and refluxed for 1.0 h. After cooling to room temperature, the mixture was vacuum filtered and washed to remove unreacted components to generate a solid precipitate in 70% isolated yield. Second, 1.2 g of the as-synthesized sample and a mixture of 1.0 mL of concentrated nitric acid and 0.5 mL of H$_2$O were added to 10 mL of HAc. The mixture was stirred at 100 °C for 1 h and then poured into

100 mL of H$_2$O after it was cooled. The precipitate was collected, washed, and dried, affording 0.41 g of BBQPH as a brown powder in 35% isolated yield. BBQPD as a reference was prepared based on a simple reaction as shown in Supplementary Fig. 3.

### Material characterizations

Fourier transform infrared (FT-IR) spectra were recorded with a Nicolet iS50. Raman spectra were collected through an inVia Reflex Evolution confocal microscope (excitation wavelength of 532 nm). X-ray diffraction (XRD) tests were performed to investigate the crystallinity of BBQPH on an Empyrean X-ray diffraction system by scanning in the 2θ range of 5–80°. X-ray photoelectron spectroscopy (XPS) characterizations were performed on a scanning X-ray microprobe (PHI5000Versa Probe ESCALAB 250xi). The calibration of the binding energy was performed at 284.8 eV relative to C1s. The morphology and microstructure were characterized by field-emission scanning electron microscopy (SEM, Zeiss Gemini-SEM450) equipped with energy-dispersive spectroscopy (EDS) for elemental analysis. High-resolution transmission electron microscopy (HRTEM) observations were carried out on a JEM-2100F. UV–vis spectrum characterization was performed on a Lambda950 in a range of 200–1200 nm to investigate the structural stability of the electrodes. Nitrogen adsorption/desorption isotherms were collected on a JW-BK200C surface area analyser operating at −196 °C to obtain the pore size of BBQPH.

### Electrode preparation and measurements

The BBQPD- and BBQPH-based electrodes for the general tests were prepared by mixing 70 wt% active materials, 20 wt% conductive carbon black (TIMCAL) and 10 wt% binders (polytetrafluoroethylene (PTFE emulsion, 10 wt% in water) or poly(1,1-difluoroethylene) (PVDF, 10 wt% in N-methylpyrrolidone (NMP)) (Sinopharm Chemical Reagent Co., Ltd.). The BBQPH/PTFE electrode was prepared by rolling and pressing it onto nickel foam (Φ = 12 mm), while the BBQPH/PVDF electrode was prepared by spreading the stock on carbon cloth (Φ = 12 mm). Then, the electrodes were dried at 60 °C for 12 h under vacuum. An active material loading of 1.5 mg cm$^{-2}$ was achieved. The zinc foil (0.03 mm thick) was sanded to remove the oxide layer on the surface and then cut into a round sheet with a diameter of 16 mm as the zinc anode. Subsequently, CR-2032 coin cell batteries were assembled with the prepared electrode as the cathode, glass fiber as the separator and Zn sheet as the counter electrode, and then, 100 μL of 3 M ZnSO$_4$ (0.3 M ZnTFS/ACN) was used as the electrolyte. In the fabrication of the Zn// BBQPH pouch cell, the prepared BBQPH powder was directly applied as the active cathode material. The active cathode materials were mixed with conductive carbon black and PVDF in an agate mortar at a ratio of 7:2:1 using N-methyl-2-pyrrolidone (NMP) as the solvent. The above slurry was then scratched onto a piece of carbon cloth (3 cm × 4 cm). The mass loading of the active material was 2.54 mg cm$^{-2}$. The typical weight of the Zn foil was 13 mg cm$^{-2}$, and its thickness was ~30 μm. The electrolyte was a 3 M ZnSO$_4$ aqueous solution. Then, the cathode, glass fiber (applied as the separator, Whatman, GF/A), and anode Zn foil were assembled into a pouch cell for further investigation.

The electrochemical performance of BBQPD and BBQPH was evaluated by galvanostatic charge/discharge (GCD) tests on a LAND CT2001A at 25 °C, and cyclic voltammetry (CV) curves and electrochemical impedance spectroscopy (EIS) spectra were collected with an electrochemical workstation (CHI 660E) within a potential of 0.2–1.6 V and a frequency range of 10$^6$-0.01 Hz, respectively. In the three-electrode system, the counter electrode was a Pt wire, and the reference electrode was Hg/HgCl. All the current densities and specific capacities of the battery were based on the active mass of the cathode.

## Density functional theory calculations and dynamics simulation

The molecular orbital levels, including the highest occupied molecular orbital (HOMO) and the lowest unoccupied molecular orbital (LUMO) levels, and the electrostatic potential (ESP) of BBQPH/BBQPD were studied at the B3LYP-D3/TZVP level of theory via the Dmol3 suite of programs. A negative (blue color) ESP suggests electrophilic properties, while a positive (red color) ESP denotes nucleophilic properties. Reduced density gradient (RDG) simulations were carried out by using the Multiwfn program, where the RDG value provides the interaction strength, while the sign($l_2$)$r$ value shows the interaction types[42]. Large and negative values of sign($l_2$)$r$ are suggestive of H-bond interactions, and values near zero indicate π-π stacking interactions[43]. The colored gradient isosurface map intuitively exhibits the interaction area and corresponding strength. VMD software was employed to plot the color-filled isosurface graphs of molecular orbitals.

The charge-storage mechanisms of BBQPD and BBQPH were investigated by using VASP. The SDD basis set was used for the Zn/H atom, and 6−31 G(d, p) was chosen for the C, H, O, and N atoms. Frequency calculations with the same methods and basis sets were performed to confirm a true local minimum and obtain the thermochemical data at P = 1 atm and T = 298.15 K. To obtain theoretical insight into the charge storage process in BBQPH, density functional theory (DFT) calculations were carried out by using the Vienna ab initio simulation package (VASP)[44]. The van der Waals (vdW) correction was considered by applying the DFT-D2 approach[45]. Projector-augmented wave (PAW) pseudopotentials were adopted with an energy cutoff of 500 eV[46]. The generalized gradient approximation with the Perdew-Burke-Ernzerhof functional (GGA-PBE) was applied to describe the exchange-correlation functional. Frequency calculations were performed with the same methods and basis sets to confirm a true local minimum and obtain the thermochemical data at P = 1 atm and T = 298.15 K. The Brillouin zone was sampled with Gamma (Γ)-centered Monkhorst-Pack mesh sampling (1 × 1 × 1 for all isolated molecular structures, 2 × 2 × 2 for all bulk crystal structures) for geometry relaxation. The convergence criteria of the residual Hellmann-Feynman force and energy during structure optimization of BBQPH were set to 0.02 eV Å$^{-1}$ and 10$^{-6}$ eV, respectively. BBQPH structures were represented by a supercell with two periodic units. Transition states were searched for by the climbing image nudged-elastic-band (CI-NEB) method, with convergence to 0.03 eV/Å and < 1 × 10$^{-7}$ eV[47]. In addition, the visualization of the structure was performed by the VESTA package.

Computational fluid dynamics (CFD) simulation was performed using COMSOL Multiphysics software to untangle the fundamental mechanisms of the Zn$^{2+}$ ion transport process in BBQPH. A flower-shaped model was constructed, which denotes the theoretical structure of BBQPH. The geometric dimensions of BBQPH were obtained from Fig. 1d. The ion diffusion process within BBQPH can be calculated and simulated according to the Poisson–Nernst–Planck equations:

$$\nabla \cdot \left( D\nabla c_i + \frac{Dz_i e}{Tk_B}\nabla\nabla c_i \right) = 0 \tag{1}$$

where D, c, z, e, T, and kB represent the diffusion coefficient (10$^{-9}$ cm$^2$ s$^{-1}$), Zn$^{2+}$ concentration (3 mol/L), Zn$^{2+}$ valence (2), elementary charge (1.6 × 10$^{-19}$ C), absolute temperature (300 K), and Boltzmann constant (1.38 × 10$^{-23}$ J/K), respectively. The mentioned equations were solved in a time-dependent state to achieve a steady state. In this work, the densest conventional triangular meshes were utilized for all simulations on the surface to pursue high accuracy of the calculation result. The MUMPS solver was used with a relative tolerance of 0.001. The volume fraction was normalized by the maximum volume fraction in the center of the BBQPH model.

## Evaluation of the energy density

According to the simplified pouch-cell configuration shown in Fig. 3h, the gravimetric energy density can be calculated based on the parameters of the assembled pouch-cell battery using the following equation:

$$E_g = \frac{VC}{\sum m_i} \tag{2}$$

where $E_g$ is the full-cell gravimetric energy density (Wh kg$^{-1}$), $V$ represents the average output voltage of the Zn//BBQPH battery (1.08 V is assumed), $C$ represents the areal capacity (mAh cm$^{-2}$), and $m_i$ is the mass per unit square (mg cm$^{-2}$) of the pouch-cell components, including the anode (5 × Zn excess is assumed) and cathode. The energy density of the pouch cell can be calculated based on actual measurements. Therefore, the energy density of the pouch cell based on active materials (BBQPH) reaches 330 Wh kg$^{-1}$, which is higher than that in most literature reports on organic materials in ZOBs. Moreover, the energy density of the pouch cell based on whole devices is -24 Wh kg$^{-1}$, which is mainly attributed to the high mass ratio of the zinc anode, diaphragm and electrolyte.

## Data availability

The experiment data that support the findings of this study are available from the corresponding authors upon reasonable request. Source data are provided with this paper.

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

## Acknowledgements

W.L. and H.X. contributed equally to this work. This work was financially supported by the National Natural Science Foundation of China (Grant No. 51773062 and 61831021). The authors also thank ECNU Multifunctional Platform for Innovation (004) for material characterizations. We also thank Xinyi Zhang for supporting us in testing and analysis.

## Author contributions

S.L. and W.L. proposed and designed the idea. W.L. performed the experimental synthesis and conducted the electrochemical measurements. H.X. performed the DFT calculation. H.Z., L.H., and S.K. assisted with the data analysis and characterization. W.L. wrote the paper. F.W. contributed to the discussion and electrochemical measurements. J.F., C.J., and J.C. contributed to the discussion. All the authors analyzed the results and commented on the manuscript.

## Competing interests

The authors declare no competing interests.
