## [Peer Review File · Nature Communications]

Steering electron delocalization of hydrogen-bonded organic framework for high-energy-density zinc-organic batteriesReviewers' comments:

Reviewer #1 (Remarks to the Author):

In this manuscript, the authors designed full-conjugated 2D hydrogen-bonded organic framework (F-HOF) as the active material of aqueous ZIBs. The F-HOF exhibits reversible Zn²⁺/H⁺ synergistic storage accompanied by 10-electron transfer. Based on such energy storage mechanism, the aqueous ZIBs deliver a specific capacity of 498.6 at 0.2 A g⁻¹. However, the work is not novel or significant enough for publication in Nat. Commun. And the detailed information is listed as following:

1. There are many works about the organic compounds with C=O and C=N active sites as the active materials in aqueous ZIBs. Moreover, the F-HOF exhibits similar energy storage mechanism with previous reports of Zn||organic batteries. The authors could give more discussion of the advantages of this system compared with other previous reports.
2. The authors compared the discharge voltage and capacity of various Zn||organic batteries in Figure 3f, where the voltage of the Zn||F-HOF battery is ~1.05 V. However, according to the GCD curve in Figure 3b, the average discharge voltage is only ~0.8 V. The authors could check this result.
3. A pouch cell was assembled in this work. The author could calculate the energy density of the pouch cell based on active materials and whole devices, respectively.
4. In Figure 5f, the Zn||F-HOF battery exhibits a capacity of ~400 mAh g⁻¹ in Zn-based organic electrolyte. There is also much capacity contribution after discharging below 1.0 V. Thus, it could not indicate that the energy storage mechanism is attributed to proton intercalation below 1.0 V.
5. The F-HOF cathode shows similar three pairs of redox peaks compared with the case in 0.05 M H₂SO₄ electrolyte (Figure 5e), indicating the similar energy storage mechanism in these two electrolytes. Why the authors indicate that the higher redox peaks are ascribed to Zn²⁺ corporation?
6. The capacitive contribution exceeds the CV curves of the Zn||F-HOF batteries (Figure 4d). It could be discussed deeply.

Reviewer #2 (Remarks to the Author):

In this manuscript, the authors developed a new type of super electron-delocalized full-conjugated 2D hydrogen-bonded organic framework (F-HOF) by steering strong nucleophilic functionality strategy for high-energy-density Zn-ions batteries. The well-designed F-HOF with super electron delocalization could simultaneously improve the redox potential, intrinsic electronic conductivity, redox kinetics, and extended energy density, thus leading to long cycling stability, high reversible capacity and rate performance. Moreover, the authors systematically investigated the redox kinetics and charge-storage mechanism of the F-HOF, revealing a boosted reversible Zn²⁺/H⁺ synergistic storage behavior accompanied by 10-electron transfer. Therefore, this work can provide new insights into the molecular designing of redox-reversible multielectron organic cathodes and bring practical prospects for high

performance. Therefore, I recommend that this paper be accepted for publication after revisions along the lines outlined below.

1. As the authors stated that the enhanced Zn²⁺ ions storage of BBQPH was attributed to the strong hydrogen bonding interactions between adjacent organics, the reviewer suggests performing more measurements to explain its excellent structural stability (e.g., thermogravimetric, UV spectrum).
2. In order to confirm that the hydrogen bond organic framework possesses faster reaction kinetics, the GITT measurement and ion diffusion coefficients of the BBQPD electrode should be provided.
3. A more thorough discussion on the mechanism section of the Zn ions battery should be provided for a better understanding of the charge storage, including ex-situ XPS and XRD analysis.
4. Why use 3 M ZnSO₄ as the electrolyte? Does the electrolyte concentration affect the stability of the BBQPH electrode during the charge/discharge process?
5. The authors should provide more information on the fabrication details of pouch batteries in the experimental section.
6. Equivalent circuit fitting should be added in Figure 5b.
7. In Supplementary Figure 25, the quantitation of the element proportion according to EDS mapping to verify the capacity contribution of protons is insufficient. The elemental analyzer is recommended to be performed for the BBQPH electrode after full discharge.

Reviewer #3 (Remarks to the Author):

In this manuscript, a long conjugate organic (benzo[a]benzo [7,8] quinoxalino[2,3-i] phenazine-5,6,8,14,15,17-hexane (BBQPH)) as the cathode for aqueous Zn organic batteries was prepared. Benefiting from the multiple C=O and C=N electroactive centers, the BBQPH cathode delivered an ultrahigh specific capacity of 498.627 mAh g⁻¹ at 0.2 A g⁻¹. Furthermore, the Zn²⁺/H⁺ synergistic storage electrochemical energy storage mechanism with the redox of BBQPH was also explored. However, some problems exist on the section of material structure and mechanism exploration, and more evidence and discussion should be provided.

1. The synthesis process shows that some by-products will be synthesized, how does the author eliminate the by-products to achieve uniform stability of the structure?
2. A key problem in the article is that the author described the synthesized material as a hydrogen-bonded organic framework, but the XRD data did not show the existence of large pores in the framework, and the article did not have a specific process to induce the assembly of the hydrogen-bonded organic framework.
3. To verify the simulated layer spacing, the TEM data of BBQPH should be given.
4. How about the cells were rested at 100 % stage followed by full discharging?
5. In Figure 3h, the corresponding charge and discharge curve of pouch-type Zn//BBQPH batteries at 0.1 A g⁻¹ should be given.
6. In Figure 3, the author does not fully display the electrochemical data. Firstly, In Figures 3d, 3e, 3g and 3h, only discharge data (I speculate) was provided, and the author does not explain whether the data is discharging capacity or charge capacity. Secondly, in Figure 3g, the author showed the electrochemical performance at three temperatures, while only one Coulombic efficiency curve was given for the three temperatures. Finally, in Figure 3h, the corresponding Coulombic efficiency should also be given.
7. For the pouch-type Zn//BBQPH batteries in Figure 3h, the author should give the

corresponding experimental details, including the amount of zinc anode, the surface loading of the cathode, etc.

8. For the mechanism exploration part, the author proposes that the high-voltage part corresponds to the intercalation of zinc ions and then the intercalation of hydrogen ions, but from the GITT curve of the material in Figure 4a, the two parts do not show differences in ion transport kinetics. Please explain the reason.

9. In Figure 5c, the XPS peaks positions of O-H, O..Zn, N-H and N..Zn is wrong, the data needs to be reanalyzed.

10. In Figure 5f, the Zn//BBQPH batteries exhibit a capacity of 400 mAh g⁻¹ using 0.5 M Zn(OTf)₂/ACN electrolyte, while the capacity contribution of Zn²⁺ is 45%, explaining the reason.

11. The CV curves of Zn//BBQPH batteries in 0.5 M Zn(OTf)₂/ACN electrolyte should be given.

12. The content of the article description Figure 2 cannot be matched with the text, the serial number in the article is wrong, please unify it.

Manuscript Number: NCOMMS-23-16480

Title: Steering Electron Delocalization of Full-Conjugated 2D Hydrogen-Bonded Organic Framework for High-Energy-Density Zn-Organic Batteries

Dear Editors and Reviewers:

We are appreciating the comments and suggestions from reviewers, and have made corresponding corrections, which would greatly improve the quality of the manuscript. We sincerely hope that you find our responses and modifications satisfactory. Please see our detailed point-by-point answers to the reviewers' comments as follows:

Reviewer #1 (Remarks to the Author): In this manuscript, the authors designed full-conjugated 2D hydrogen-bonded organic framework (F-HOF) as the active material of aqueous ZIBs. The F-HOF exhibits reversible Zn^{2+}/H^+ synergistic storage accompanied by 10-electron transfer. Based on such energy storage mechanism, the aqueous ZIBs deliver a specific capacity of 498.6 at 0.2 $A\ g^{-1}$. However, the work is not novel or significant enough for publication in Nat. Commun. And the detailed information is listed as following:

Response: We are appreciating for your comments. In this work, we developed a full-conjugated 2D hydrogen-bonded organic framework (F-HOF) with super electron delocalization based on theoretical prediction and guidance for molecular designing. The F-HOF has the highest density of active sites as Zn-organic cathode among the ever reported (*Adv. Mater.* 2021, 33, 2106469; *J. Am. Chem. Soc.* 2021, 143, 15369-15377; *Angew. Chem.* 2020, 132, 18478-18489; *ACS Energy Lett.* 2021, 6, 11411147). In addition, this work also demonstrated the novelty in electron density regulation of redox-active structures, in-depth elaboration of the super electron delocalization effect, and the in-depth systematic theoretical/experimental mechanisms analysis. And it not only presents a new HOF with excellent electrochemical properties, but also provides a new avenue for designing high-performance organic frameworks for energy storage. The detail novelty and significations of this work are as follows:

(i) A full-conjugated 2D hydrogen-bonded organic framework (F-HOF) with super electron delocalization was developed based on theoretical prediction and guidance. Different from the common conjugated organic configuration, **the unique super electron delocalization**

simultaneously improved the redox potential, intrinsic electronic conductivity, and redox kinetics. Besides, the 2D hydrogen-bonded organic framework, benzo[a]benzo [7,8] quinoxalino[2,3-i] phenazine-5,6,8,14,15,17-hexane (BBPDH), **featuring synergistic C=O and C=N electron-withdrawing motifs** that alter the intramolecular electron distribution, thus **boost the redox voltage and trigger multi-electron storage chemistry for ZOBs.** Notably, **the intermolecular multiple hydrogen bonds (C=O \cdots H/C=N \cdots H) together with the π - π stacking interactions further enhanced the structural stability of BBQPH.**

(ii) Profiting from the synergistic advantages aroused from the intrinsic strongly polarized electron distribution and tight intermolecular non-covalent locking, the ZOBs with BBQPH electrodes delivered **a high output voltage of 1.2 V, record-breaking capacity (498.6 mAh g⁻¹ at 0.2 A g⁻¹), corresponding to a best-in-class energy density of 355 Wh kg⁻¹, and outstanding cycling performance of >1000 cycles at 5.0 A g⁻¹ for aqueous ZOBs.** Noteworthy the BBQPH represents one of **the best comprehensive performances surpassing most reported organic-based electrodes.** Notably, **even at a wide work temperature range,** a high capacity of 295 mA h g⁻¹ is still retained. Moreover, the fabricated **Zn-ion pouch cell output a 14 mAh capacity** at a current density of 0.1 A g⁻¹, and the satisfactory energy density of the battery reflects the excellent designed concept.

(iii) Computational fluid dynamics simulation revealed the structural advantages of superior ion diffusion and storage capability at the mesoscopic level. Furthermore, the reaction pathway accompanied by 10-electron transfer for the 2D HOF-based cathode has also been disclosed by a series of *ex-situ* investigations and theoretical simulations. Meanwhile, **the intermolecular nanochannels facilitate the desolvation of hydrated Zn²⁺ ions and boost fast Zn²⁺ migration with the low diffusion barrier (1.42 eV),** affording BBQPH with outstanding rate capability and extraordinary stability during long-term cycling.

We believe that the novelty and significance of this work will arouse widespread interest among researchers. Moreover, these modifications from you and other reviewer's suggestions will greatly increase the novelty and quality of the work. Therefore, we sincerely appreciate the comments and suggestions from you, and hope you find our responses and modifications satisfactory. Then, please see our point-by-point answers as below:

1. There are many works about the organic compounds with C=O and C=N active sites as the active materials in aqueous ZIBs. Moreover, the F-HOF exhibits similar energy storage mechanism with previous reports of Zn||organic batteries. The authors could give more discussion of the advantages of this system compared with other previous reports.

Response: Thank you for this feedback and suggestions. We provided more discussion of the advantages of this work accordingly. Although various *many works about the organic compounds with C=O and C=N active sites as the active materials in aqueous ZIBs*. The recently reported conjugated systems by molecule modifications and conjugate structure extension strategy often involve insufficient redox activities of functional groups that inevitably reduce the capacity (*Nat. Commun. 2021, 12(1): 4424; Adv. Mater. 2022, 2207115; Angew. Chem. 2023, e202219136; Energy Environ. Sci., 2023, 16, 89*). Besides, common organic molecules are often subjected to dissolution in the electrolyte during the redox process and thereby suffer from severe shuttle effects and poor cyclability. Till now, there is still a lack of an ideal organic cathode to realize high-energy-density and long-cycle life ZOBs.

Different from the previous literature in which by simple introducing functional groups for regulating the potential. Here, **we developed a super electron-delocalized full-conjugated 2D hydrogen-bonded organic framework (F-HOF) by steering a strong nucleophilic functionality strategy** for high-energy-density ZOBs. Concretely, the super electron delocalization alters the intramolecular electron distribution thus significantly boosting the redox potential. Meanwhile, the intermolecular multiple hydrogen bonds endowed it present transverse 2D extension and longitudinal π - π stacking structure, which extended the electron delocalization area, narrowed the energy gap between HOMO and LUMO, and boosted structural stability. **Benefiting from these synergetic superiorities, the F-HOF delivered an elevated output voltage, an ultrahigh reversible capacity of 498.6 mAh g⁻¹ at 0.2 A g⁻¹, outstanding cyclability (capacity retention of 95 % after 1000 cycles) and high energy density (355 Wh kg⁻¹) for ZOBs. Besides, the fabricated Zn-ion pouch cell output a 14.0 mAh capacity at a current density of 0.1 A g⁻¹. Moreover, the redox kinetics and charge-storage mechanism of the F-HOF were systematically investigated, revealing a boosted reversible Zn²⁺/H⁺ synergistic storage behavior accompanied by 10-electron transfer.**

This work offers new insights into the molecular designing of redox-reversible multielectron

organic cathode and bright practical prospects for high performances ZOBs. These achievements are expected to have a great impact on other research of high energy density metal-organic batteries, which combined high-quality results in organic cathode designing, clarified $\text{Zn}^{2+}/\text{H}^+$ synergistic storage chemistry mechanism, and emerging energy storage applications. We hope that our manuscript will be of great interest to researchers in chemistry, energy storage, and other related disciplines.

2. The authors compared the discharge voltage and capacity of various Zn//organic batteries in Figure 3f, where the voltage of the Zn//F-HOF battery is ~ 1.05 V. However, according to the GCD curve in Figure 3b, the average discharge voltage is only ~ 0.8 V. The authors could check this result.

Response: Thanks for your suggestion. We carefully checked the average discharge voltage of the Zn//F-HOF battery in Figure 3b and Figure 3f accordingly. In order to demonstrate the outstanding performance of the Zn//F-HOF battery more clearly, the average discharge voltage of the Zn//F-HOF battery was marked in the corresponding GCD curve in Figure 3b.

Figure 3. b) Galvanostatic discharge/charge curves of the BBQPH and BBQPD electrodes at 0.2 A g^{-1} .

Figure 3. f) Comparison of capacity and energy density between BBQPH electrode and recently reported organic cathodes.

3. A pouch cell was assembled in this work. The author could calculate the energy density of the pouch cell based on active materials and whole devices, respectively.

Response: Thanks for your professional suggestion. The energy density of the pouch cell based on active materials and whole devices was calculated and provided in the Supporting Information section accordingly.

The evaluation of energy densities: On the basis of the simplified pouch-cell configuration shown in Figure 3h, the E_g can be evaluated the basis of pouch-cell data using the equations:

$$E_g = \frac{VC}{\sum m_i}$$

where E_g is the full-cell gravimetric (Wh kg^{-1}) energy densities, V is the average output voltage (1.08 V is assumed), C is the areal capacity (mAh cm^{-2}), m_i is the mass per unit square (mg cm^{-2}) of cell components including the cathode, anode ($2\times\text{Zn}$ excess is assumed), the energy densities of pouch cells are calculated based on actual measurements. Therefore, the energy density of the pouch cell based on active materials (BBQPH) is reach to 330 Wh kg^{-1} that close to the energy density of a coin cell batteries (355 Wh kg^{-1}), which higher most literature reports on organic materials in ZOBs. Moreover, the energy density of the pouch cell based on whole devices is about 24 Wh kg^{-1} due to the high mass ratio of diaphragm and electrolyte. Accordingly, developing lighter diaphragms and employ less electrolyte to increase the energy density of devices is the focus of our next research work.

4. In Figure 5f, the Zn//F-HOF battery exhibits a capacity of $\sim 400 \text{ mAh g}^{-1}$ in Zn-based organic electrolyte. There is also much capacity contribution after discharging below 1.0 V. Thus, it could

not indicate that the energy storage mechanism is attributed to proton intercalation below 1.0 V.

Response: Thank you for your questions. The Zn//F-HOF batteries exhibit similar redox behavior in 0.5 M Zn(OTf)₂/ACN, 0.05 M H₂SO₄, and 3.0 M ZnSO₄ electrolytes according to the CV measurement (Figure 5e and Figure S24c). Therefore, The F-HOF cathode has a good ability both in to store zinc ions and proton due to its high density of active sites and strong electron delocalization properties. To unveil the contribution during discharging below 1.0 V, a series of characterizations (*ex-situ* XRD and XPS) are applied, the *ex-situ* XRD and XPS results revealed the H⁺ coordination behavior below 1.0 V. In detail, the characteristic peak of Zn₄SO₄(OH)₆·xH₂O (8.17°) derived from proton insertion gradually appears and disappears during discharging and charging, respectively (Figure 4c, Figures S22, 23). Moreover, an *in-situ* pH test was conducted using a direct-reading pH meter in a self-made two-electrode reaction device to track the pH change (Figure 5a). The pH value shows an increase during discharge (voltage range from 1.0 to 0.2 V) and a downward trend when charging, revealing the existence of reversible proton storage behaviour for F-HOF. Therefore, the capacity contribution after discharging below 1.0 V can be ascribed to H⁺ corporation. Based on this valuable comment, the manuscript has been revised accordingly.

Page 11 line 14: “Therefore, an *in-situ* pH test was conducted using a direct-reading pH meter in a self-made two-electrode reaction device to track the pH change (Figure 5a). The pH value shows an increase during discharge (voltage range from 1.0 to 0.2 V) and a downward trend when charging, revealing the existence of reversible proton storage behaviour for BBQPH.³⁶”

Page 11 line 22: “The characteristic peak of Zn₄SO₄(OH)₆·xH₂O (8.17°) derived from proton insertion gradually appears and disappears during discharging and charging, respectively (Figure 4c, Figures S22, 23). Moreover, the typical peaks of BBQPH slightly decrease occur while new peaks appear upon the cation intercalation process, corroborating the Zn²⁺/H⁺ reversible co-storage behaviour.”

Figure S27. CV curves of BBQPH at 1.0 mV s^{-1} in (a) $0.05 \text{ M H}_2\text{SO}_4$ aqueous electrolyte and (b) 3 M ZnSO_4 aqueous electrolyte measured by three-electrode systems, respectively.

Note: The dotted line represents the corresponding CV curves in $0.05 \text{ M H}_2\text{SO}_4$ solution after being shifted to imitate its acting in a dilute H_2SO_4 electrolyte with a pH value of about 3.7.

Figure S29. (a) The charging and discharging curves and (b) corresponding cycling performance of Zn/BBQPH battery using $0.5 \text{ M Zn(OTf)}_2/\text{ACN}$ electrolyte. (c) The CV curves of Zn/BBQPH battery employing $0.5 \text{ M Zn(OTf)}_2/\text{ACN}$ electrolyte

Note: The Zn/BBQPH battery using $0.5 \text{ M Zn(OTf)}_2/\text{ACN}$ electrolyte can still delivers satisfactory specific capacity at low current density (0.1 A g^{-1}). However, the battery demonstrates lower specific capacity at a high current density (5.0 A g^{-1}), which can be attributed to the slow reaction kinetics for Zn^{2+} storage. Despite this, the Zn/BBQPH battery using $0.5 \text{ M Zn(OTf)}_2/\text{ACN}$ still shows the best electrochemistry performances compared with organic material for pure zinc ion storage reported.

Charge storage mechanism:

Figure S28. The charge storage mechanism in the different electrolytes.

Figure 5. Redox chemistry mechanism. a) Schematic of the in-situ pH detection equipment and corresponding pH evolution diagram of the BBQPH cathode in redox. b) in-situ EIS spectra. c) The ex-situ high-resolution XPS spectra of N 1s and d) O 2p.

Figure S31. The optimized $\text{Zn}^{2+}/\text{H}^{+}$ co-storage reaction path of BBQPH.

Note: We adopted a para-equivalent ion insertion method due to the special center rotational symmetry of BBQPH to simplify DFT calculations.

5. The F-HOF cathode shows similar three pairs of redox peaks compared with the case in 0.05 M H_2SO_4 electrolyte (Figure 5e), indicating the similar energy storage mechanism in these two electrolytes. Why the authors indicate that the higher redox peaks are ascribed to Zn^{2+} corporation?

Response: Thank you for your questions. The Zn//BBQPH batteries exhibit similar redox behavior in 0.5 M $\text{Zn}(\text{OTf})_2/\text{ACN}$, 0.05 M H_2SO_4 , and 3.0 M ZnSO_4 electrolytes according to the CV measurement (Figure 5e and Figure S24c). Therefore, The F-HOF cathode has a good ability both in to store zinc ions and proton due to its high density of active sites and strong electron delocalization properties. Moreover, the *ex-situ* XRD and XPS further confirmed the $\text{Zn}^{2+}/\text{H}^{+}$ synergistic coordination mechanism.

To unveil the contribution of proton storage, we further investigated the electrochemical properties of the BBQPH electrode using an electrolyte without Zn salt (i.e., 0.05M H_2SO_4), and aqueous ZnSO_4 electrolytes with different concentrations, respectively (Figures S28-29). We confirmed the capacity contribution of Zn^{2+} is 45% in 3.0 M ZnSO_4 electrolyte according to the EDS mapping analysis results (Figure S30). Therefore, the higher redox peaks are ascribed to Zn^{2+} corporation.

Figure 5. Redox chemistry mechanism. a) Schematic of the *in-situ* pH detection equipment and corresponding pH evolution diagram of the BBQPH cathode in redox. b) *in-situ* EIS spectra. c) The *ex-situ* high-resolution XPS spectra of N 1s and d) O 2p. e) CV curves of BBQPH at 1.0 mV s⁻¹. f) Zn²⁺/H⁺ co-storage mechanism and possible redox processes of BBQPH at each platform.

Figure S29. (a) The charging and discharging curves and (b) corresponding cycling performance of Zn/BBQPH battery using 0.5 M Zn(OTf)₂/ACN electrolyte. (c) The CV curves of Zn/BBQPH battery employing 0.5 M Zn(OTf)₂/ACN electrolyte.

Note: The Zn/BBQPH battery using 0.5 M Zn(OTf)₂/ACN electrolyte can still deliver satisfactory specific capacity at low current density (0.1 A g⁻¹). However, the battery demonstrates lower specific capacity at a high current density (5.0 A g⁻¹), which can be attributed to the slow

reaction kinetics for Zn^{2+} storage. Despite this, the Zn/BBQPH battery using 0.5 M $\text{Zn}(\text{OTf})_2/\text{ACN}$ still shows the best electrochemistry performances compared with organic material for pure zinc ion storage reported.

Figure S28. The charge storage mechanism in the different electrolytes.

Figure S30. (a) The SEM image of BBQPH after full discharge and (b) contribution ratio of Zn^{2+} and proton of Zn/BBQPH battery using 3.0 M ZnSO_4 electrolyte. (c) EDS mapping of the BBQPH electrode after full discharge.

6. The capacitive contribution exceeds the CV curves of the Zn//F-HOF batteries (Figure 4d). It could be discussed deeply.

Response: We appreciate the suggestion of the reviewer. This phenomenon of the capacitive

contribution exceeding the CV curves originates from peak voltage offset with the increase in sweep speed (*Angew. Chem. Inter. Ed.*, 2021, 60, 21310-21318; *Angew. Chem. Inter. Ed.*, 2023, 135, 202219136). In detail, with the increase of scan rate in the CV test, the oxidation peaks would inevitably shift positively while reduction peaks move negatively due to polarization, equivalently caused by a non-ignorable resistance between potential and current. To avoid any sort of controversy, we have recalculated the capacitive contribution after correcting the polarization voltage in the Revised Manuscript.

Figure 4. d) Capacitive and diffusion contribution ratios at 1.0 mV s⁻¹.

Reviewer #2 (Remarks to the Author): In this manuscript, the authors developed a new type of super electron-delocalized full-conjugated 2D hydrogen-bonded organic framework (F-HOF) by steering strong nucleophilic functionality strategy for high-energy-density Zn-ions batteries. The well-designed F-HOF with super electron delocalization could simultaneously improve the redox potential, intrinsic electronic conductivity, redox kinetics, and extended energy density, thus leading to long cycling stability, high reversible capacity and rate performance. Moreover, the authors systematically investigated the redox kinetics and charge-storage mechanism of the F-HOF, revealing a boosted reversible Zn²⁺/H⁺ synergistic storage behavior accompanied by 10-electron transfer. Therefore, this work can provide new insights into the molecular designing of redox-reversible multielectron organic cathodes and bring practical prospects for high performance. Therefore, I recommend that this paper be accepted for publication after revisions along the lines outlined below.

Response: We appreciate the positive comments on our work, and we have revised the manuscript accordingly.

1. As the authors stated that the enhanced Zn^{2+} ions storage of BBQPH was attributed to the strong hydrogen bonding interactions between adjacent organics, the reviewer suggests performing more measurements to explain its excellent structural stability (e.g., thermogravimetric, UV spectrum).

Response: Thank you very much for your suggestions. We have performed thermogravimetry and UV spectrum measurements to explore the structural stability of the BBQPH accordingly. The thermogravimetric analysis (TGA) measurement of BBQPH indicated that BBQPH is stable even at 439 °C without obvious weight loss, which indicates the presence of multiple hydrogen bonds enhanced the thermostability of the BBQPH. In addition, the UV spectrum further revealed its excellent structural stability in water solvents. Based on this valuable comment, the manuscript has been revised accordingly.

Page 6 line 19: “It should be pointed out that the existence of strong intermolecular forces of π - π interactions and hydrogen bonding renders its high thermostability and poor solubility in water solvents (Figures S7-8), thus extremely advantageous for suppressing the shuttle effect and favorable cycle stability in aqueous batteries.^{26, 27}”

Figure S7. The TGA curve of the BBQPH material.

Figure S8. The photograph of the BBQPH immersed in water for 3 days and the corresponding UV spectrum.

2. In order to confirm that the hydrogen bond organic framework possesses faster reaction kinetics, the GITT measurement and ion diffusion coefficients of the BBQPD electrode should be provided.

Response: Thanks for your suggestion. The GITT measurement and ion diffusion coefficients of the BBQPD electrode are provided accordingly. Based on this suggestive comment, our manuscript has been revised accordingly.

Page 9 line 20: “The galvanostatic intermittent titration technique (GITT) is first carried out to explore the redox kinetics of two electrodes (**Figure 4a**). The higher ion diffusion coefficients ($1.5 \times 10^{-9} \text{ cm}^2 \text{ s}^{-1}$ vs. $6.8 \times 10^{-10} \text{ cm}^2 \text{ s}^{-1}$ for BBQPD) combined with low active energy (1.58 eV) calculated from electrochemical reaction curves evidenced the faster reaction kinetic and superior ions transport ability of BBQPH (Figures S18, 19).³²”

Figure S19. The (a) GITT curve and (b) diffusion coefficient of the BBQPD cathode.

3. A more thorough discussion on the mechanism section of the Zn ions battery should be provided for a better understanding of the charge storage, including *ex-situ* XPS and XRD analysis.

Response: Thanks for this helpful suggestion. More detailed discussions on the mechanism section of the Zn-ions battery were provided in order to have a better understanding of the charge storage accordingly.

Page 11 line 21: “Following *ex-situ* XRD, FTIR and X-ray photoelectron spectroscopy (XPS) were further performed to elucidate the chemical structure evolution of BBQPH upon the charge storage process. The characteristic peak of $\text{Zn}_4\text{SO}_4(\text{OH})_6 \cdot x\text{H}_2\text{O}$ (8.17°) derived from proton insertion gradually appears and disappears during discharging and charging, respectively (Figure 4c, Figures S22, 23). Moreover, the typical peaks of BBQPH slightly decrease occur while new peaks appear upon the cation intercalation process, corroborating the $\text{Zn}^{2+}/\text{H}^+$ reversible co-storage behaviour. In addition, the two typical peaks in FTIR spectra weaken in the reduction process, corresponding to the coordination of cations with the redox-active centers (Figure S24). Similarly, the XPS spectra of O 2p and N 1s revealed the evolution of the C=O and C=N bonds, there are transformed into the C-N bond (400.1 eV) and C-O bond (533.3 eV) upon discharge process (Figure 4d), respectively, which suggested the coordination reaction of the C=N/C=O two redox centers with $\text{Zn}^{2+}/\text{H}^+$.^{38, 39} In the subsequent oxidation process, the C=O/C=N peaks reappear and gradually strengthen, illustrating the release of $\text{Zn}^{2+}/\text{H}^+$ and the highly reversible feature of the BBQPH redox activity.”

4. Why use 3 M ZnSO_4 as the electrolyte? Does the electrolyte concentration affect the stability of the BBQPH electrode during the charge/discharge process?

Response: We thank the reviewer for carefully reviewing our manuscript and for raising these questions. According to the referee’s suggestion, we have conducted the electrochemical performance investigations of different concentrations of ZnSO_4 electrolytes.

In general, the concentration of the electrolyte has a significant effect on the electrochemical properties of the electrode materials. Among them, ionic conductivity and acidity of the electrolyte play a key role. There are exist plentiful protons in aqueous Zn^{2+} electrolytes due to the hydrolysis reaction of ZnSO_4 . Therefore, on the one hand, with the increase in the concentration of zinc sulfate electrolyte, the ionic conductivity will also increase. On the other hand, free water molecules will decrease in high concentrations in the electrolyte due to strong ionic coordination

effects, which is beneficial to reduce the solubility of organic materials in the electrolyte for enhancing cycling stability. In this work, we conducted the physics/electrochemical performance investigations of different concentrations of ZnSO₄ electrolytes. As expected, as the electrolyte concentration increases, the pH also decreases slightly. Moreover, the aqueous 3 M ZnSO₄ electrolyte shows excellent ionic conductivity (>45 mS cm⁻¹) and thus promising for Zn-organic batteries (Figure S21). For the lower concentration electrolytes (0.5, 1.0, and 2.0 M ZnSO₄), the low Coulombic efficiency and poor cycle stability make it difficult to be applied in the Zn-organic batteries (Figure S22). Therefore, the 3.0 M ZnSO₄ electrolyte has unique advantages in both performance and cost compared to other electrolytes. Based on this suggestive comment, our manuscript has been revised accordingly. Corresponding comparing results have been added in the Supplementary Information.

Page 12 line 9: “To unveil the contribution of proton storage, we further investigated the electrochemical properties of the BBQPH electrode using an electrolyte without Zn salt (i.e., 0.05M H₂SO₄), and aqueous ZnSO₄ electrolytes with different concentrations, respectively (Figures S25, 26).”

Figure S25. (a) The photographs of the 0.5, 1.0, 2.0, 3.0 and 4.0 M ZnSO₄ electrolytes. (b) The corresponding pH of the electrolytes. (c) The Coulombic efficiencies and cycling performances of Zn//BBQPH batteries using different concentrations of ZnSO₄ electrolytes.

5. The authors should provide more information on the fabrication details of pouch batteries in the experimental section.

Response: Thanks for this helpful suggestion. The details information on the fabrication of pouch

batteries is provided in the experimental section accordingly. Based on this suggestive comment, our manuscript has been revised accordingly. Corresponding fabrication details have been added in the Supplementary Information.

Page 5 line 5 in the Supporting information: “In the fabrication of Zn//BBQPH pouch cell, the prepared BBQPH powder was directly applied as the active cathode materials. The active cathode materials were mixed with conductive carbon black and PVDF in an agate mortar at a ratio of 7:2:1, using N-methyl-2-pyrrolidone (NMP) as the solvent. The above slurry was then scratched to a piece of carbon cloth (3cm×4cm). The mass loading of the active material was 2.54 mg cm⁻². The typical weight of Zn foil is 13 mg cm⁻², and its thickness was ~20 μm. The electrolyte was a 3 M ZnSO₄ aqueous solution. Then, the cathode, glass fiber (applied as the separator, Whatman, GF/A) and anode Zn foil were assembled into a pouch cell for further investigation.”

6. Equivalent circuit fitting should be added in Figure 5b.

Response: Thanks for this helpful suggestion. Corresponding equivalent circuit fitting results have been added in **Figure 5b** of the revised manuscript, as suggested.

Figure 5. Redox chemistry mechanism. b) in-situ EIS spectra.

7. In Supplementary Figure 25, the quantitation of the element proportion according to EDS mapping to verify the capacity contribution of protons is insufficient. The elemental analyzer is recommended to be performed for the BBQPH electrode after full discharge.

Response: Thank you for your suggestion. We have performed element analyze (XPS spectrum etc.) for the BBQPH electrode after full discharge accordingly. The corresponding element content of the electrode was listed in Table S1. According to the XPS spectrum result, we can conclude that the Zn²⁺ contributes about 45% of the capacity in the BBQPH electrode, while the capacity contribution of protons can reach 55%, which is consistent with the EDS mapping analysis results.

Figure S30. (a) The SEM image of BBQPH after full discharge and (b) contribution ratio of Zn^{2+} and proton of Zn/BBQPH battery using 3.0 M ZnSO_4 electrolyte. (c) EDS mapping of the BBQPH electrode after full discharge.

Table S1. The proportion of elemental mass after discharge in the BBQPH electrode based on XPS spectral results.

Element	Wt. %
C	11.3
N	1.5
O	14.2
S	4.1
Zn	68.9

Reviewer #3 (Remarks to the Author): In this manuscript, a long conjugate organic (benzo[a]benzo [7,8] quinoxalino[2,3-i] phenazine-5,6,8,14,15,17-hexane (BBQPH)) as the cathode for aqueous Zn organic batteries was prepared. Benefiting from the multiple C=O and C=N electroactive centers, the BBQPH cathode delivered an ultrahigh specific capacity of 498.627 mAh g^{-1} at 0.2 A g^{-1} . Furthermore, the $\text{Zn}^{2+}/\text{H}^+$ synergistic storage electrochemical energy storage mechanism with the redox of BBQPH was also explored. However, some problems exist on the section of material structure and mechanism exploration, and more evidence and

discussion should be provided.

Response: We are appreciating for the comments from the reviewer. We have provided more evidences and discussion on the section of material structure and mechanism exploration accordingly.

1. The synthesis process shows that some by-products will be synthesized, how does the author eliminate the by-products to achieve uniform stability of the structure?

Response: We appreciate the professional question of the reviewer. Admittedly, the synthesis of BBQPH hydrogen-bonded organic framework involves the generation of by-products in this work. It is noted that the by-product exists in another molecular configuration, which results in the by-product cannot be assembled through hydrogen bonds, further allowing them to dissolve in the reaction solvent during the synthesis process. While the BBQPH molecules can well spontaneously self-assemble forming a stable hydrogen-bonded organic framework that can be easily separated from the reaction system. Therefore, the synthesis strategy provides convenience for eliminating the by-products to achieve uniform stability of the BBQPH hydrogen-bonded organic framework structure. In addition, we performed the ^1H NMR measurement to further dispel this concern (Figure S2), which verified the successful synthesis of BBQPH with high purity. Based on this suggestive comment, our manuscript has been revised accordingly.

Page 5 line 14: “The ^1H NMR spectra verified the successful synthesis of BBQPH (Figure S2). Fourier transform infrared spectroscopy (FTIR) of BBQPH showed that the peaks located at 1590 and 1689 cm^{-1} attributed to the C=N and C=O tensile vibration groups, respectively.²³ The Raman spectra further definitely confirmed the molecular functional structures of the two products as expected (Figure S3).²⁴”

Figure S2. The ^1H NMR spectroscopy of prepared BBQPH.

2. A key problem in the article is that the author described the synthesized material as a hydrogen-bonded organic framework, but the XRD data did not show the existence of large pores in the framework, and the article did not have a specific process to induce the assembly of the hydrogen-bonded organic framework.

Response: Thanks for your professional question. The assembly of the BBQPH hydrogen-bonded organic framework is implemented in tandem with the synthesis in this work. Specifically, we introduced to water and acetic acid in the synthesis process (Scheme S2), which could provide a favorable environment for the BBQPH to achieve assembly (*Nat. Commun.* **2020**, *11*, 3180; *Angew. Chem. Int. Ed.* **2022**, *61*, e202116289, etc.). Notedly that the BBQPH molecule shows a rotationally symmetric molecular configuration consisting of multiple carbonyls and pyrazine groups, which could be facile to form a fully conjugated 2D hydrogen-bonded organic framework by intermolecular non-covalent locking. Owing to the multi-site hydrogen bond interactions, the resultant F-HOF is highly robust. In addition, relevant structural characterization (XRD, HRTEM, etc.) and theoretical simulation further confirmed the successful preparation of HOFs (Figure S5). Therefore, the strategy of simultaneous synthesis and assembly for obtaining HOF materials is available. Meanwhile, the strategy also provides a new avenue for designing high-performance organic frameworks for energy storage.

For the XRD part, the resultant F-HOF demonstrated a tight arrangement due to its unique molecular configuration, which lead to a smaller stacked pore structure (0.54 nm) and high compaction density (Figure S10), which makes the diffraction peak corresponding to the structural

hole insignificant in XRD data. It is noted that this phenomenon is very common especially in microporous HOF materials (*Angew. Chem. Int. Ed.* 2023, 135, e202217710; *Adv. Mater.* 2021, 33, 2106079, etc.). In order further confirmed the hydrogen-bonded organic framework structure, we performed the HRTEM measurement for the BBQPH, the High-resolution TEM image visualized the ordered and robust structure of the BBQPH materials with lattice fringes of 0.33 nm reveals, which is consistent with the XRD measurement. Therefore, the hydrogen-bonded organic framework (F-HOF) is successfully constructed in this work.

Page 6 line 15: “the High-resolution TEM image visualized the ordered and robust structure of the BBQPH materials with lattice fringes of 0.33 nm reveals, which is consistent with the XRD measurement.”

Scheme S2. Synthetic route of BBQPH.

Figure 2. Structural characterization. a) PXRD pattern of BBQPH framework. b) Stacking diagram with a packing distance of 3.3 Å together with C=O...H/C=N...H non-covalent locking models of BBQPH framework. c) Plot of density gradient decreasing with sign (λ_2) ρ . d) The SEM image of modulated BBQPH crystal. e) EPR of the BBQPH.

Figure S6. High-resolution transmission electron microscopy (HRTEM) image of BBQPH.

Figure S10. (a) N₂ adsorption/desorption isotherms of the BBQPH. (b) The micropores size distribution curves by t-plot analyses. (c) The mesopores size distribution curves by BJH analyses.

Note: The BBQPH demonstrates a specific surface area of $68.12 \text{ m}^2 \text{ g}^{-1}$ combined with a multistage pore size of 0.54 nm and 18 nm. The t-Plot results show that the micropores (0.54 nm) exist inside the BBQPH hydrogen-bonded organic framework. While the mesoporous (18 nm) arises from the accumulation of sponge structures.

Figure S5. The calculated crystal structure of BBQPH.

Note: The full-conjugated structure and the rich C=O and C=N groups of BBQPH molecule make it connected to four adjacent BBQPH molecules by multiple hydrogen bonding between weak hydrogen donor C-H groups and strong hydrogen bond acceptor C=O/C=N groups (C=O \cdots H/C=N \cdots H), forming 2D planar supramolecular structure. The bond lengths of the hydrogen bond between the hydrogen atom in C-H and the oxygen atom in the carbonyl group were about 2.35 Å and 2.40 Å, which are consistent with the typical hydrogen bonds in this work.

3. To verify the simulated layer spacing, the TEM data of BBQPH should be given.

Response: Thanks for your valuable suggestion. The TEM measurement was performed to verify the simulated layer spacing of the BBQPH accordingly. Based on this important suggestion, our manuscript has been revised accordingly. Corresponding TEM measurement results have been added in the Supplementary Information.

Page 6 line 15: “the High-resolution TEM image visualized the ordered and robust structure of the BBQPH materials with lattice fringes of 0.33 nm reveals, which is consistent with the XRD measurement.”

Figure S6. High-resolution transmission electron microscopy (TEM) image of BBQPH.

4. How about the cells were rested at 100 % stage followed by full discharging?

Response: The reviewer makes a good point. We performed rested at 100 % stage followed by full discharging accordingly. Correspondingly, a more thorough discussion has been added in the revised manuscript.

Page 9 line 30: “Considering the surface-controlled charge storage character, the anti-self-discharge and anti-dissolution stability are conducted that further revealing its excellent capacity retention and structural stability (Figure S20). Accordingly, the dominating reason responsible for the ultrafast charge storage is attributed to strong electron delocalization brings highly electroactive C=O/C=N zincophilic sites and robust molecular H-bonded organic network, which facilitates faster redox kinetics, enhanced stability, and elevated conductivity than that BBQPD (Figure 4f).”

Figure S20. Electrode stability studies. (a) The Zn/BBQPH cells in 3 M ZnSO₄ were first fully charged to 1.6 V at 1.0 A g⁻¹ (based on active materials of cathode), and then the cells were rested at 50 % stage of charge (SOC) for 3.0 h, followed by full discharging. (b) The Zn/BBQPH battery was first fully charged to 1.6 V at 2.0 A g⁻¹, and then the cells were rested at 100 % stage of charge (SOC) for 3.0 h, followed by full discharging. (c) The photograph of the disassembled cell after 1000 cycles. (d) The photographs of the electrode immersed in the electrolyte at different times. (e) UV-visible spectroscopy of TABQ. (f) ex-situ UV-visible spectroscopy of electrolytes in Zn/BBQPH cells during the charge/discharge process.

5. In Figure 3h, the corresponding charge and discharge curve of pouch-type Zn//BBQPH batteries at 0.1 A g⁻¹ should be given.

Response: Thanks for your good suggestion. The corresponding charge and discharge curve of pouch-type Zn//BBQPH battery at 0.1 A g⁻¹ is provided accordingly.

Page 9 line 12: “Moreover, the assembled Zn-ion pouch cell output a 14.0 mAh capacity at 0.1 A g⁻¹ (Figure 3h and Figure S17), the such exceptional electrochemical performance of the battery

holds considerable promise in practical application and highlights the advantages of this molecular design, including increased energy density and reduced solubility of organic materials.”

Figure S17. The charge and discharge curves of pouch-type Zn//BBQPH battery at 0.1 A g^{-1} .

6. In Figure 3, the author does not fully display the electrochemical data.

Firstly, In Figures 3d, 3e, 3g and 3h, only discharge data (I speculate) was provided, and the author does not explain whether the data is discharging capacity or charge capacity. Secondly, in Figure 3g, the author showed the electrochemical performance at three temperatures, while only one Coulombic efficiency curve was given for the three temperatures. Finally, in Figure 3h, the corresponding Coulombic efficiency should also be given.

Response: Thank you very much for your suggestions. We redraw the Figures 3d, 3e, 3g and 3h to fully display the electrochemical data accordingly.

Page 8 line 3: “Specifically, the BBQPH electrode shows superior rate performance with energy density and capacities of 498.6, 455.0, 436.8, 422.9, 406.2, and 393.6 mAh g^{-1} at 0.2, 0.5, 1.0, 2.0, 5.0, and 8.0 A g^{-1} , respectively (Figure 3d, Figures S13,14). Moreover, when the current density is returned to 0.2 A g^{-1} , the capacity gets recovered to 487.2 mAh g^{-1} , unravelling the excellent rate performances and reversibility of the BBQPH electrode. For the BBQPD electrode, the corresponding capacities monotonically decreased from 305.4 to 208.3 mAh g^{-1} as the applied current density increased from 0.2 to 8.0 A g^{-1} .”

Figure 3. Electrochemical performances of the BBQPH and BBQPD cathode. a) The contrastive CV curves of BBQPH and BBQPD cathode at 0.5 mV s^{-1} . b) Galvanostatic discharge/charge curves of the BBQPH and BBQPD electrodes at 0.2 A g^{-1} . c) Bar graph of the discharge plateau versus the energy density. d) Rate performances and e) cycling stability for BBQPH and BBQPD. f) Comparison of capacity and energy density between BBQPH electrode and recently reported organic cathodes. g) Cycling performances of BBQPH electrodes at the different test temperatures. h) The cycling performance of pouch-type Zn//BBQPH batteries at 0.1 A g^{-1} .

7. For the pouch-type Zn//BBQPH batteries in Figure 3h, the author should give the corresponding experimental details, including the amount of zinc anode, the surface loading of the cathode, etc.

Response: Thank you very much for your suggestions. The experimental details of the pouch-type Zn//BBQPH batteries are provided accordingly. Correspondingly, a more thorough experimental details has been added in the Supporting Information.

Page 5 line 5 in the Supporting information: “In the fabrication of Zn//BBQPH pouch cell, the

prepared BBQPH powder was directly applied as the active cathode materials. The active cathode materials were mixed with conductive carbon black and PVDF in an agate mortar at a ratio of 7:2:1, using N-methyl-2-pyrrolidone (NMP) as the solvent. The above slurry was then scratched to a piece of carbon cloth (3cm×4cm). The mass loading of the active material was 2.54 mg cm⁻². The typical weight of Zn foil is 13 mg cm⁻², and its thickness was ~20 μm. The electrolyte was a 3 M ZnSO₄ aqueous solution. Then, the cathode, glass fiber (applied as the separator, Whatman, GF/A) and anode Zn foil were assembled into a pouch cell for further investigation.”

8. *For the mechanism exploration part, the author proposes that the high-voltage part corresponds to the intercalation of zinc ions and then the intercalation of hydrogen ions, but from the GITT curve of the material in Figure 4a, the two parts do not show differences in ion transport kinetics. Please explain the reason.*

Response: Thanks for your professional suggestion. Generally, the GITT test consists of a series of "pulse/constant current/relaxation" that can well determine the chemical diffusion coefficient. The noticeable differences in transport kinetics of BBQPH in the intercalation process of zinc ions and hydrogen ions can be attributed to the following reasons. On the one hand, due to the unique structural properties of BBQPH molecules, rendering both zinc-ions intercalation and protons intercalation process exhibit very fast reaction kinetics. This results in an insignificant difference in reaction kinetics from GITT. On the other hand, in general, the proton intercalation process has faster reaction kinetics. However, due to the increase of steric hindrance as ion intercalation progresses together with the overall Gibbs free energy does not change much, which slows down the reaction kinetics of the proton intercalation process and shows a kinetic similar to that of the zinc intercalation process. This can also explain this common phenomenon that the same ion embedded in different voltage intervals exhibits different reaction kinetics (*Angew. Chem. Int. Ed. 2023, 135(9): e202217710; Adv. Mater. 2022, 2207115; Angew. Chem. Int. Ed. 2022, 134, e202115180; Energy Environ. Sci. 2020, 13, 2515-2523*). Therefore, the GITT showing similar ion transport kinetics during the charge/discharge process is reasonable and valid.

9. *In Figure 5c, the XPS peaks positions of O-H, O...Zn, N-H and N...Zn is wrong, the data needs to be reanalyzed.*

Response: Thank you for your valuable suggestion. We have refitted the XPS spectra after correcting and reanalyzed the corresponding XPS peak positions accordingly. Meanwhile, the

refitted XPS spectra agree with previous reports (*Angew. Chem. Int. Ed.* 2023, e202219136; *Adv. Mater.* 2022, 2207115; *Angew. Chem. Int. Ed.* 2022, 61, e202116289; *Energy Storage Mater.* 2021,40, 31-40).

Page 11 line 27: “Similarly, the XPS spectra of O 2p and N 1s revealed the evolution of the C=O and C=N bonds, there are transformed into the C-N bond (400.1 eV) and C-O bond (533.3 eV) upon discharge process (Figure 4d), respectively, which suggested the coordination reaction of the C=N/C=O two redox centers with Zn^{2+}/H^+ .^{38, 39} In the subsequent oxidation process, the C=O/C=N peaks reappear and gradually strengthen, illustrating the release of Zn^{2+}/H^+ and the highly reversible feature of the BBQPH redox activity.”

Figure 5 c) The ex-situ high-resolution XPS spectra of N 1s and d) O 2p.

10. In Figure 5f, the Zn//BBQPH batteries exhibit a capacity of 400 mAh g⁻¹ using 0.5 M Zn(OTf)₂/ACN electrolyte, while the capacity contribution of Zn^{2+} is 45%, explaining the reason.

Response: We thank the reviewer for carefully reviewing our manuscript and for raising these questions. Admittedly, the Zn//BBQPH batteries exhibit a capacity of 400 mAh g⁻¹ using 0.5 M Zn(OTf)₂/ACN electrolyte, indicating the BBQPH electrode has a good ability to store zinc ions due to its high density of active sites and strong electron delocalization properties. It is noted that the BBQPH electrode also demonstrates excellent proton storage capacity with faster reaction kinetics according to the CV measurement (Figure 5e). In addition, there are exist plentiful protons in aqueous Zn^{2+} electrolytes (3.0 M ZnSO₄) due to the hydrolysis reaction of Zn salt. Therefore, there has a strong tendency for Zn^{2+} and H^+ cations to synergistically coordinated with the phenanthroline (C=N) and C=O groups. Moreover, the *ex-situ* XRD and XPS further confirmed the Zn^{2+}/H^+ synergistic coordination mechanism.

To unveil the contribution of proton storage, we further investigated the electrochemical

properties of the BBQPH electrode using an electrolyte without Zn salt (i.e., 0.05M H₂SO₄), and aqueous ZnSO₄ electrolytes with different concentrations, respectively (Figures S28-29). We confirmed the capacity contribution of Zn²⁺ is 45% in 3.0 M ZnSO₄ electrolyte according to the EDS mapping analysis results (Figure S30).

Page 12 line 9: “To unveil the contribution of proton storage, we further investigated the electrochemical properties of the BBQPH electrode using an electrolyte without Zn salt (i.e., 0.05M H₂SO₄), and aqueous ZnSO₄ electrolytes with different concentrations, respectively (Figures S25, 26). Notably, the BBQPH cathode demonstrated similar three pairs of redox peaks with lower redox potentials in 0.05M H₂SO₄ electrolyte according to calibrated CV curves (Figure 6e), indicating the initial coordination is derived from Zn²⁺ and followed by continuous two successive H⁺ uptake. Besides, a non-aqueous Zn²⁺ electrolyte (0.5 M Zn(OTf)₂/ACN) was adopted to further confirm the priority coordination of Zn²⁺ and exclude the effect of protons (Figures S27, 28), the higher Zn²⁺-coordination potential further confirmation of previous speculation. Interestingly, the BBQPH electrode still exhibits moderately reduced capacity (372 mAh g⁻¹) but with lower redox kinetics (Figure S29). In the view above analyses, the Zn²⁺ contributes about 45% of the capacity in the BBQPH electrode, while the capacity contribution of protons can reach 55%, which is consistent with the EDS mapping analysis and XPS analysis results (Figure S30, Table S1). Based on the above mechanism studies, it is clear that the carbonyl and phenazine groups of BBQPH have highly reversible redox activity, and the stepwise co-intercalation of Zn²⁺/H⁺ enables it to exhibit excellent redox dynamics.”

11. The CV curves of Zn//BBQPH batteries in 0.5 M Zn(OTf)₂/ACN electrolyte should be given.

Response: Thanks for your valuable suggestion. We have conducted CV measurement of Zn//BBQPH batteries in 0.5 M Zn(OTf)₂/ACN electrolyte accordingly.

Page 12 line 14: “Besides, a non-aqueous Zn²⁺ electrolyte (0.5 M Zn(OTf)₂/ACN) was adopted to further confirm the priority coordination of Zn²⁺ and exclude the effect of protons (Figures S27, 28), the higher Zn²⁺-coordination potential further confirmation of previous speculation. Interestingly, the BBQPH electrode still exhibits moderately reduced capacity (372 mAh g⁻¹) but with lower redox kinetics (Figure S29).”

Figure S29. (a) The charging and discharging curves and (b) corresponding cycling performance of Zn/BBQPH battery using 0.5 M Zn(OTf)₂/ACN electrolyte. (c) The CV curves of Zn/BBQPH battery employing 0.5 M Zn(OTf)₂/ACN electrolyte.

Note: The Zn/BBQPH battery using 0.5 M Zn(OTf)₂/ACN electrolyte can still delivers satisfactory specific capacity at low current density (0.1 A g⁻¹). However, the battery demonstrates lower specific capacity at a high current density (5.0 A g⁻¹), which can be attributed to the slow reaction kinetics for Zn²⁺ storage. Despite this, the Zn/BBQPH battery using 0.5 M Zn(OTf)₂/ACN still shows the best electrochemistry performances compared with organic material for pure zinc ion storage reported.

12. The content of the article description Figure 2 cannot be matched with the text, the serial number in the article is wrong, please unify it.

Response: Thank the reviewer for carefully reviewing our manuscript and raising this very valuable advice. We have corrected this error in the Revised Manuscript accordingly.

REVIEWERS' COMMENTS

Reviewer #1 (Remarks to the Author):

Authors have revised the manuscript significantly. However, there are many works about the organic compounds with C=O and C=N active sites as the active materials in aqueous ZIBs. Moreover, the F-HOF exhibits similar energy storage mechanism with previous reports of Zn||organic batteries. Therefore, I still think that the work is not novel or significant enough for publication in Nat. Commun.

Reviewer #2 (Remarks to the Author):

In this revised manuscript, the authors have thoroughly responded the comments and suggestions of reviewers, and provided more experimental data and characterizations, thus remarkably improving the novelty and quality of the work. Especially, considering the advances in organic cathode designing, storage mechanism understanding, and high-energy-density ZIB performance, I recommend the acceptance of this work in Nat. Comm.

Reviewer #3 (Remarks to the Author):

The paper has been well improved and now it can be accepted as it is.

Responses to the comments of the reviewers

We are appreciating the comments and suggestions from reviewers, and have made corresponding corrections, which greatly improved the quality of the manuscript. We have addressed or clarified the issues raised in the reviewers' reports, and sincerely hope that you find our responses and modifications satisfactory. Please see our detailed point-by-point answers to the reviewers' comments as follows:

REVIEWERS' COMMENTS

Reviewer #1 (Remarks to the Author):

Authors have revised the manuscript significantly. However, there are many works about the organic compounds with C=O and C=N active sites as the active materials in aqueous ZIBs. Moreover, the F-HOF exhibits similar energy storage mechanism with previous reports of Zn||organic batteries. Therefore, I still think that the work is not novel or significant enough for publication in Nat. Commun.

Response: We thank the reviewer for carefully reviewing our manuscript and raising this point. Here we explained the signification and novelty of our work accordingly, and sincerely hope that you find our responses and modifications satisfactory.

Generally, the electrochemical performances of aqueous ZIBs mainly focus on the energy density, power density and cycle performances (*Angew. Chem. Int. Ed.* 2020, 59, 21293-21303; *Nat. Commun.* 2021, 12(1): 4424; *ACS Energy Lett.* 2018, 3, 2480-250). Indeed, many organic compounds containing C=O and C=N active functional groups have been reported in the literature for zinc-ion batteries, but many of them are classified as small molecular electrode materials, which always exhibit high solubility in electrolytes and then result in the shuttle of active materials and fast capacity decay in batteries (*Adv. Funct. Mater.* 2023: 2306675; *ACS Nano* 2023, 17, 3, 3077-3087). In addition, the imine (C=N) compounds often have a low average discharge voltage, which limits their practical application (*Adv. Mater.* 2022, 2207115; *Angew. Chem. Int. Ed.* 2020, 132, 4950-4954; *Nano-Micro Lett.* 2023, 15, 36). In this work, we developed a full-conjugated 2D hydrogen-bonded organic framework (F-HOF) with super electron delocalization based on theoretical prediction and molecular designing guidance. Notably, different from the previous

literature in which by simply introducing functional groups for regulating the potential, the quinone (C=O) and phenazine (C=N) of F-HOF not only as multi-active centers but also served as polar groups with electron donors to induce the formation of hydrogen bonding. Therefore, compared with the small organic compounds, the hydrogen-bonded organic framework displays lower solubility in water due to their strong intermolecular hydrogen bonding connected structures, which endows aqueous ZOBs with long cycle stability (*Angew. Chem.* 2023, e202219136). Moreover, this innovative molecular design strategy alters the intramolecular electron distribution thus significantly boosting the redox potential. Furthermore, the extended electron delocalization area narrowed the energy gap between HOMO and LUMO, endowing a faster charge transfer of the F-HOF. It is worth mentioning that the designed F-HOF exhibited **the highest density of active sites as Zn-organic cathode among the ever reported** (Figure 3f) (*Adv. Mater.* 2023, 35(22): 2301088; *J. Am. Chem. Soc.* 2021, 143, 15369-15377; *Angew. Chem. Int. Ed.* 2020, 132, 18478-18489; *ACS Energy Lett.* 2021, 6, 11411147). **Benefiting from these synergetic superiorities, the F-HOF delivered an elevated output voltage, an ultrahigh reversible capacity of 498.6 mAh g⁻¹ at 0.2 A g⁻¹, outstanding cyclability (capacity retention of 95 % after 1000 cycles) and high energy density (355 Wh kg⁻¹) for ZOBs.**

The redox kinetics and charge-storage mechanism of the F-HOF were systematically investigated in this work, revealing a boosted reversible Zn²⁺/H⁺ synergistic storage behavior accompanied by 10-electron transfer. More importantly, we firstly clarified the charge storage step for C=O and C=N motifs of F-HOF during the redox process based on a series of *ex-situ* characterizations and theoretical calculations, which broadened the horizons of aqueous ZOBs chemistry. Additionally, we further investigated the effects of hydrogen bond networks on charge storage and ion diffusion. The Zn²⁺ mobility within the F-HOF was first computed using the DFT model to figure out the optimal ion migration path (Figure 6d). Two representative migration paths were evaluated based on a transition state search. Profiting from the unique hydrogen bond network structure, the lower migration energy barrier (1.42 eV) within molecular-constructed nanochannels promotes rapid penetration of Zn²⁺/H⁺ into the internal active sites and boosting fast redox dynamics, thus affording F-HOF to realize outstanding rate performances as well as superb stability during long-term cycling.

Therefore, this work clearly demonstrated the signification of molecular engineering strategy

based on electron density regulation of redox-active structures, in-depth elaboration of the super electron delocalization effect, and the in-depth systematic theoretical/experimental mechanisms analysis. And it not only presents a new HOF with excellent electrochemical properties, but also provides a new avenue for designing high-performance organic frameworks for energy storage.

Fig. 3f Comparison of capacity and energy density between BBQPH electrode and recently reported organic cathodes.

Fig. 6d Schematic demonstrating the Zn^{2+} migration along the P1 path and corresponding migration barrier curve.

Reviewer #2 (Remarks to the Author):

In this revised manuscript, the authors have thoroughly responded the comments and suggestions of reviewers, and provided more experimental data and characterizations, thus remarkably improving the novelty and quality of the work. Especially, considering the advances in organic cathode designing, storage mechanism understanding, and high-energy-density ZIB performance, I recommend the acceptance of this work in Nat. Commun.

Response: We highly appreciate your positive evaluation. Thank you for the recommendation to publish our work.

Reviewer #3 (Remarks to the Author):

The paper has been well improved and now it can be accepted as it is.

Response: We thank the reviewer for the careful review of our manuscript and the recommendation to publish this work.